



# An active power control approach for wake-induced load alleviation in a fully developed wind farm boundary layer

Mehdi Vali[1], Vlaho Petrović[1], Gerald Steinfeld[1], Lucy Y. Pao[2], and Martin Kühn[1]

[1]ForWind - Center for Wind Energy Research, Institute of Physics, University of Oldenburg, 26129 Oldenburg, Germany
[2]Department of Electrical, Computer & Energy Engineering, University of Colorado Boulder, USA

*Correspondence to:* Mehdi Vali (mehdi.vali@uni-oldenburg.de)

**Abstract.** This paper studies a closed-loop wind farm control framework for active power control (APC) with a simultaneous reduction of wake-induced structural loads within a fully developed wind farm flow interacting with the atmospheric boundary layer. The main focus is on a classical feedback control, which features a simple control architecture and a practical measurement system that are realizable for real-time control of large wind farms. We demonstrate that the wake-induced structural loadings of the downstream turbines can be alleviated, while the wind farm power production follows a reference signal. A closed-loop APC is designed first to improve the power tracking performance against wake-induced power losses of the downwind turbines. Then, the non-unique solution of APC for the wind farm is exploited for aggregated structural load alleviation. The axial induction factors of the individual wind turbines are considered as control inputs to limit the power production of the wind farm or to switch to greedy control when the demand exceeds the power available in the wind. Furthermore, the APC solution domain is enlarged by an adjustment of the power set-points according to the locally available power at the waked wind turbines. Therefore, the controllability of the wind turbines is improved for rejecting the intensified load fluctuations inside the wake. A large-eddy simulation model is employed for resolving the turbulent flow, the wake structures and its interaction with the atmospheric boundary layer. The applicability and key features of the controller are discussed with a wind farm example consisting of 3×4 turbines with different wake interactions at each row. The performance of the proposed APC is evaluated using the accuracy of the wind farm power tracking and the wake-induced damage equivalent fatigue loads of the towers of the individual wind turbines.

## 1 Introduction

The number and size of large-scale wind farms are rapidly growing worldwide due to more deployment of wind energy. Wind Europe has proposed wind energy targets to produce almost half of the European Union (EU) electricity demand by 2050. One pathway to reach this ambitious goal is further improvement of the cost-effectiveness of existing wind farms, e.g., by optimal control of the overall wind farm performance. Control of turbines in a wind farm is challenging because of the aerodynamic interactions via wakes. The characteristics of a wake are reduced wind speed and increased turbulence. The former diminishes the total power production of the farm and the latter leads to a higher dynamic loading on the downstream turbines. In a wind farm, wakes often merge with each other, resulting in the so-called multiple wakes. The wake interactions also depend strongly



on variations of wind speed and direction, the surface roughness, turbulence, different atmospheric stratifications, local terrain, and turbine layout in a wind farm (Barthelmie et al., 2009).

Traditionally each turbine operates greedily to locally capture the maximal amount of kinetic energy of the incoming wind without considering its aerodynamic impact on its downwind turbines. By changing operating points of upwind turbines, it
is possible to influence their wakes, and as a result the performance of downwind turbines, possibly increasing their energy extraction or decreasing their structural fatigue loads. The main objectives of wind farm control can be categorized as follows (Knudsen et al., 2015; de Alegria et al., 2007)

1. Power maximization by e.g., minimizing the wake-induced power losses,

2. Active power control (APC), wherein the total wind farm power production follows a power reference signal provided
by the transmission system operator (TSO),

3. Aggregated structural load alleviation, e.g., by coordinating the power distributions among the individual wind turbines or by mitigating the wake-induced dynamic loads,

4. Voltage and frequency control of wind farms for connecting to the grid.

The idea to maximize the power production of wind farms in the presence of wakes is to coordinate the control settings of
the individual wind turbines, by taking their wake interactions into account. Two commonly proposed approaches for wake control of wind farms are induction control and wake-steering control. In the first method, the upwind turbines reduce their own power productions, the so-called derated operation, to increase the amount of available kinetic energy of the wakes reaching downstream turbines (Schepers and van der Pijl, 2007; Annoni et al., 2016). In the latter method, the rotors of upwind machines are intentionally yawed out of the wind direction in order to deflect their own wakes away from their downwind
turbines (Wagenaar et al. 2012; Gebraad et al., 2016).

Furthermore, future wind farms should be able to stabilize the grid frequencies through control of their power production in order to balance power supply with demand, the so-called active power control (APC). Aho et al. (2012, 2016) have investigated providing APC services at the wind turbine level. Fleming et al. (2016) have studied APC for a wind farm plant and demonstrate the challenge of high wake conditions when the overall power reference is distributed evenly among the wind turbines. For
such a power reference distribution, highly waked turbines can adversely degrade the grid reliability due to lack of enough wind power in reserve. Indeed, they generate below the power command due to the wakes and thus enlarge the total wind farm power tracking error.

The increased turbulence intensity inside the wakes, wake meandering, and partial wake overlapping of the rotors of downwind turbines cause waked turbines to be prone to higher structural fatigue loading. Several studies have utilized optimization
techniques to find optimal set-points in order to simultaneously maximize the total power production and prolong the lifespan of the wind farm through minimizing the structural fatigue loading (van Dijk et al., 2017; Kanev et al., 2018). Nonetheless, from a control engineering perspective, these have been either open-loop or quasi-steady-state optimization approaches, based on analytical static wake models and data-driven load models, which do not fully hold for dynamical wake interactions. Fur-





thermore, there still exists a research gap for extending the trend of the active load control at the wind turbine level (Schlipf et al., 2013; Vali et al., 2016) to the wind farm level.

Recently, several studies have developed model predictive control for both power maximization (Goit and Meyers, 2015; Munters and Meyers, 2018; Vali et al., 2018a) and active power control (Shapiro et al., 2017; Vali et al., 2018b) in wind farms by taking the wake interactions into account. A fully 3D large-eddy simulation (LES) model is utilized in Goit and Meyers (2015) and Munters and Meyers (2018) to optimize the overall wind farm performance. However, MPC schemes proposed in Vali et al. (2018a), Shapiro et al. (2017), and Vali et al. (2018b) rely on simplified wind farm models for capturing the dominant dynamic wake interactions in a computationally efficient manner. It has been shown that adequate MPC formulations and parameterizations can significantly reduce the computational complexity (Vali et al., 2018a), however, the overall complexity of such a control system, including a suitable mathematical model (Boersma et al., 2018; Rott et al., 2017) and measurement system (Doekemeijer et al., 2018), is still an open research topic, particularly for large wind farms. Distributed MPC has recently received attention in order to reduce the computational burden of model predictive APC (Bay et al., 2018).

Soleimanzadeh et al. (2013) and Madjidian (2016) have studied optimal control for structural fatigue load alleviation of wind farms, though the performance of the controllers is assessed using simplified engineering models. To obtain a more comprehensive understanding of the underlying physics, and how control techniques can influence dynamic wind farm flows, computational fluid dynamics (CFD) models tuned with experimental data, have been typically employed. These models allow resolving time-varying turbulent flow for better characterization of the wake physics, e.g., wake meandering, wake-added turbulence, shape of the velocity deficit inside the wake, multiple wakes, different atmospheric stabilities, etc.

On the other hand, model-free and classical control approaches have also received attention due to their simple control architecture and ease of implementation for real-time control of large wind farms (Marden et al., 2013; Gebraad and van Wingerden, 2015). Both allow the performance of the designed controllers to be evaluated with more realistic wind farm flow conditions, e.g., free field testing (Fleming et al., 2017), wind tunnel testing (Campagnolo et al., 2016; Petrović et al., 2018), and high-fidelity LES models (van Wingerden et al., 2017; Ciri et al., 2017; Vali et al., 2018c). Extremum seeking control has been studied as a closed-loop realization of an optimization problem for power maximization of waked wind farms. However, slow convergence time, sensitivity to changes in atmospheric conditions, and high turbulence conditions significantly affect the performance of extremum seeking controllers (Johnson and Fritsch, 2012; Vali et al., 2017).

Classical control approaches have demonstrated good potential for practical APC of waked wind farms to stabilize wind power penetrations into the grid. van Wingerden et al. (2017) have proposed a classical feedback loop using the total wake-induced power tracking error to improve the grid stability independent from the selection of the wind turbine power set-points. However, the pattern of structural loadings of the wind turbine components changes for a different distribution of the set-points and despite smooth tracking of the total power, the fluctuations of powers and associated loads of the individual turbines can increase significantly. The non-unique solution of APC for wind farms has been exploited in Vali et al. (2018c) for influencing the wind turbine structural loadings, while their total power production follows a demanded power reference from the TSO. That study focused only on the cases wherein the APC solution domain is large enough for adjusting the wind turbine power set-points, leading to a better dynamic load balance of all wind turbines inside a waked wind farm. However, a high TSO





power demand mainly limits the APC solution domain to the upwind turbines because of their highest amount of wind power in reserve. In such conditions, the upwind turbines might be loaded even higher than the waked turbines to compensate the tremendously accumulated wake-induced wind farm power tracking errors. Therefore, a load balance within a wind farm may cause sacrificing the accuracy of the tracking performance or even instability.

The main contribution of this paper is an extension to the APC approaches proposed in Fleming et al. (2016), van Wingerden et al. (2017), and Vali et al. (2018c) to actively coordinate the power set-points of the individual wind turbines for reducing the structural fatigue loadings in a waked wind farm. The classical feedback control architecture enables individual wind turbine control systems to cooperatively realize the wind farm control objectives using only practically available local power and structural load measurements. The model-free nature of the proposed control approach makes it prominent for real-time control

of a waked wind farm independent from its size and wake model complexities. Therefore, large-eddy simulations (LES) are utilized for testing the closed-loop control performance under detailed dynamic wake and turbulent conditions. The LES approach resolves the unsteady nature of the wake, e.g., intensified turbulent fluctuations, wake recoveries, and wake meandering within the wind farm, while the wind turbines are represented by a dynamic actuator disc model (ADM). The constraints of the control problem subjected to different levels of demands and wake losses are investigated for a reliable APC and grid stability.

We discuss how the power set-points of the individual wind turbines should be adjusted to enlarge the APC solution domain in high-waking situations. Three main features of the wind farm control approach presented in this paper, the so-called APC with coordinated load distribution (CLD), are:

1. An effective switching between the APC and the greedy control, when the total available wind power in reserve is lower than the demanded power reference.

2. An effective usage of the locally available wind power at the waked turbines to enlarge the APC solution domain and thus increase the controllability of the wind turbines for rejecting the intensified structural loadings inside the wakes.

3. Avoiding inefficient loop interactions between the two designed control loops, i.e., APC and CLD loops, when the control solution domain is so limited.

The remainder of this paper is organized as follows. In section 2, we briefly present the large-eddy simulation model used,

the employed wind turbine model, and the wind farm layout used for the simulations in this investigation. The main focus of section 3 is on the control architectures for APC at both the wind turbine and the wind farm levels. The constraints of the control problem are also presented. A test simulation scenario for APC verification is introduced in section 4. Then, the performance of the proposed APC with CLD is discussed through comparative large-eddy simulation studies. The performances are evaluated with two criteria, the wind farm power tracking accuracy and the fatigue load distribution. The corresponding wake-induced

structural loadings of the individual wind turbines are discussed as well. Finally, the strengths and weaknesses of the proposed approach are outlined in section 5 as conclusions of the current study.



## 2 Wind farm simulation model

The background atmospheric boundary layer, especially in the first rows of a wind farm, has a huge impact on the characteristics of the wake recovery, wherein a mixing process with the turbulent inflow re-energizes the wake (Sanderse, 2009). The LES approach has shown capability of resolving the unsteady nature of the wake and turbulent flows within a wind farm (Meyers and Meneveau, 2010; Porté-Agel et al., 2011; Sarlak, 2014).

### 2.1 The PArallelized LES Model (PALM)

In this study, an LES wind farm model is utilized, which employs the PArallelized Large-eddy simulation Model (PALM) version 4.0 (Maronga et al., 2015) coupled with the Actuator Disc Model (ADM) of a wind turbine (Witha et al., 2014). PALM is an open source LES code, which is developed for simulating atmospheric and oceanic flows and optimized for massively parallel computer architectures. PALM makes use of the Schumann volume averaging approach (Schumann, 1975) and uses central differences to discretize the non-hydrostatic and incompressible Boussinesq approximation of the three-dimensional Navier-Stokes equations on a structured Cartesian grid.

Much more detailed wind turbine models with more realistic near wake structure, e.g., the actuator disc model with rotation (ADM-R) and actuator line model (ALM) are also implemented in PALM. However, ADM is computationally efficient and provides a good approximation of the far wake structure, making it suitable for the present study. The static ADM is extended with a first-order lag and a mass-spring-damper system in order to estimate the dominant aerodynamic power response and the first tower fore-aft mode excitations of an individual wind turbine to the wind farm turbulent flow and wakes in a computationally efficient manner.

### 2.2 The wind turbine model

The individual wind turbines are parameterized with an actuator disc model (ADM) to exert a thrust force on the incoming flow and extract a certain amount of energy from the incoming wind. The wind turbines are modeled in PALM with only two degrees of freedom (DoF) as follows

$$\tau \dot{P} + P = P_a(a, U_0, \dot{x}_T) \tag{1}$$

$$m_{T_e}\ddot{x}_T + c_T\dot{x}_T + k_T x_T = F_a(a, U_0, \dot{x}_T) \tag{2}$$

Equation (1) represents the power response of a wind turbine to the aerodynamic power $P_a$ with the aerodynamic time constant $\tau$, which can be associated to the drive-train dynamics. The axial induction factor $a$ stands for the ratio of the reduced wind velocity at the rotor to the effective wind speed $U_0$ at a far distance upwind from the rotor disc and can be translated to the practical torque and pitch control inputs of a wind turbine. Equation (2) describes the first tower fore-aft dynamic mode with the tower top fore-aft displacement $x_T$, the aerodynamic thrust force $F_a$, and the tower equivalent modal mass $m_{T_e}$, structural damping $c_T$ and bending stiffness $k_T$ according to Schlipf et al. (2013). Note that the tower top fore-aft displacement $x_T$ is considered positive in the downwind direction.

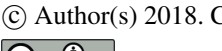



The aerodynamic thrust force for a single turbine is calculated using the employed ADM in the PALM simulation code as follows (Gasch and Twele, 2011):

$$F_a = \frac{1}{2}\rho A_d U_{rel}^2 C_T(a), \tag{3}$$

where the thrust coefficient $C_T$ is described as a function of the axial induction factor $a$ as:

$$C_T(a) = 4a(1-a), \tag{4}$$

and $\rho$ is the air density and $A_d$ is the swept area of the rotor plane. Note that the axial induction factor is limited to its value for maximum (greedy) power extraction (Gasch and Twele, 2011), i.e., $a \leq \frac{1}{3}$, to avoid high wake losses and violating the thrust approximation at higher induction factors. The relative wind speed $U_{rel}$ is defined as a superposition of the effective wind speed and the structural tower top velocity as (Schlipf et al., 2013):

$$U_{rel} = U_0 - \dot{x}_T, \tag{5}$$

in order to model the aerodynamic damping of the tower fore-aft mode.

Resolving equation (5) and considering the induction effect of a rotor disc, the effective wind speed $U_0$ can be approximated from the measurable axial disc-averaged wind velocity $U_d$ from the PALM code as

$$\hat{U}_0 = \frac{U_d}{1-a} + \dot{x}_T, \tag{6}$$

and enables us to model the applied aerodynamic thrust force (3) acting in negative direction on the flow (Witha et al., 2014). Then, the aerodynamic power of an individual wind turbine is calculated as

$$P_a = F_a \cdot U_d, \tag{7}$$

The tower base fore-aft bending moment of an individual wind turbine is approximated as (Schlipf et al., 2013):

$$M_{yT} = h_H (c_T \dot{x}_T + k_T x_T), \tag{8}$$

where $h_H$ is the hub height. In this investigation wind turbines are operating in the below-rated region, wherein the effective wind speed $U_0$ is always below the rated wind speed of the simulated wind turbine. It is assumed that the generator torque is fast enough to realize the commanded axial induction factor. Therefore, the actuator dynamics are neglected here.

Figure 1 plots the power response of the simulated wind turbine model (1)-(2), which is incorporated later in the PALM code for simulating a wind farm example and used for the controller designs in the following sections. The power response of the
model is compared with the resultant electrical power of the first order drive-train dynamics presented in (Schlipf et al., 2013) without considering any electromechanical losses. Table 1 lists the key parameters of the wind turbine models, which are taken from the freely available model of the NREL 5MW reference wind turbine (Jonkman and Buhl, 2005).

Both models in Fig. 1 are subjected to the same effective wind velocities at the optimal operating point of the NREL 5MW reference turbine with rotor diameter $D = 126\,\text{m}$, which is obtained from steady-state power and thrust computations as a



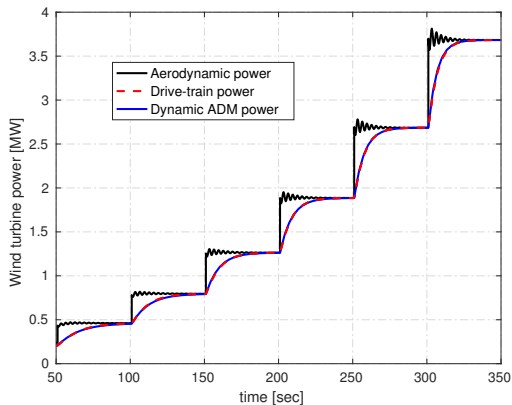

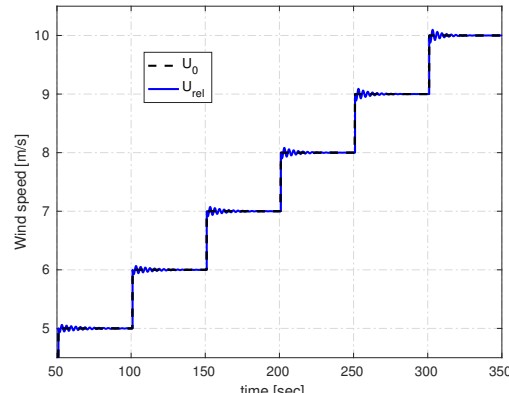

**Figure 1.** Power response of the simulated ADM compared with the resultant electrical power output of the drive-train model. Both models are subjected to the same wind speeds.

**Figure 2.** Ramped effective wind speed $U_0$ and the relative wind velocity $U_{rel}$, which takes the tower top fore-aft movement into account according to (5).

**Table 1.** Specifications of the used wind turbine models taken from Jonkman and Buhl (2005).

| Simulation parameter | value |
|---|---|
| Turbine rotor diameter $D$ | 126 m |
| Hub height | 90 m |
| Rated wind speed $U_{rated}$ | 11.2 m/s |
| Aerodynamic time constant $\tau$ | 8 s* |
| Tower equivalent modal mass $m_{T_e}$ | 436,865 kg |
| Tower equivalent structural damping $c_T$ | 17,568 kg/s |
| Tower equivalent bending stiffness $k_T$ | 1,766,242 kg/s$^2$ |
| Operational region | below-rated |
| Simulation sample time | 0.01 s |

* Time constant at an effective wind speed $U_0 = 8$ m/s.

function of the effective wind speed, e.g., using blade element momentum (BEM) theory. The aerodynamic time constant $\tau$ in (1) is scheduled as a function of the effective wind speed in order to describe the resultant electrical power of the considered drive-train as a reference. The fluctuations of the calculated aerodynamic power (black curves) originate from tower fore-aft mode excitations because of the step changes of the wind speed $U_0$. Note that the above analysis on the wind turbine model has

5   not been simulated with PALM as it is not possible to step the wind speeds in the PALM code. Figure 2 compares the ramped effective and the relative velocities, showing the effect of the tower structural dynamic model (2) on the aerodynamic power and thrust of the extended ADM. The following section focuses on our investigated wind farm example, which is simulated after coupling the presented wind turbine model (1)-(2) with PALM (see level 0 of Fig. 6).





## 2.3 The case study

A layout of a 3×4 wind farm example with different wake overlaps with downstream wind turbines is considered here (Figure 3). The wind turbines are spaced $5D$ in the stream-wise direction. In the first and third rows, i.e., turbines $\{1, 4, 7, 10\}$ and $\{3, 6, 9, 12\}$, the rotor centers of the downstream turbines are intentionally offset half a rotor diameter from the centers of their

upwind ones, while the wind turbines in the middle row, i.e., turbines $\{2, 5, 8, 11\}$, are spaced without lateral offset to create full wake interactions. Figure 3 shows the instantaneous field of the $u$-component of the wind at the hub height of the wind turbines. Our convention for the rows ($R_1$ to $R_3$) and columns ($C_1$ to $C_4$) of wind turbines is also clarified. Table 2 summarizes the key parameters of the PALM simulation set-up for the main simulations of the considered wind farm case study with active power control.

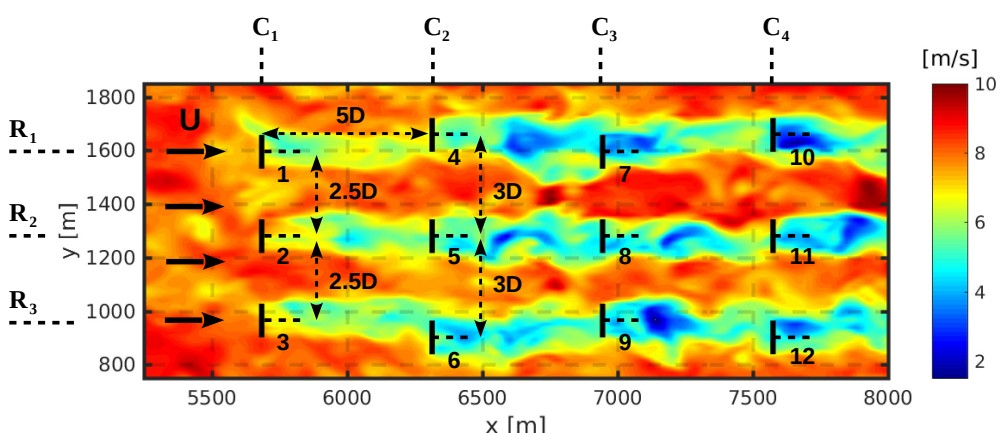

**Figure 3.** The layout of the waked 3×4 wind farm model simulated with PALM. $R_i$ and $C_j$ stand for the $i^{th}$ row and the $j^{th}$ column of the simulated wind turbines, respectively.

A neutral boundary layer (NBL) is simulated with a capping inversion and a mean wind speed of 8 m/s at hub height. Under such conditions, large wake losses can be expected during standard operation of the wind farm, i.e., without active power control. A precursor simulation of the atmospheric boundary layer is conducted without any turbines in order to allow for the generation of a fully developed undisturbed turbulent flow field. Then, it is used for the initialization of the main wind farm simulation runs in which a turbulence recycling method is used.

A high turbulence intensity is one of the key drivers for fatigue loading under both ambient and wake conditions (Frandsen, 2007). After resolving wake structures and its propagation through the wind farm, the longitudinal turbulence intensities of two arbitrary points inside the free-stream and the wake are calculated. In this study, the upwind turbines are subjected to free stream flow with the longlitudinal turbulence intensity of approximately 5% at hub height, whereas the downstream turbines operate inside wakes with longitudinal turbulence intensity of about 15% in the wake center. Note that the turbulence intensity

will indeed be slightly higher, as generally the subgrid-scale turbulent kinetic energy (TKE) will also contribute. Furthermore,



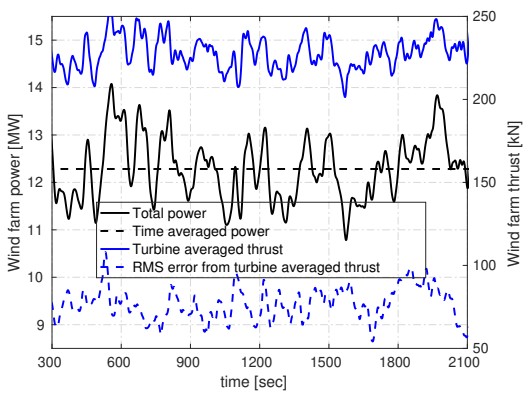

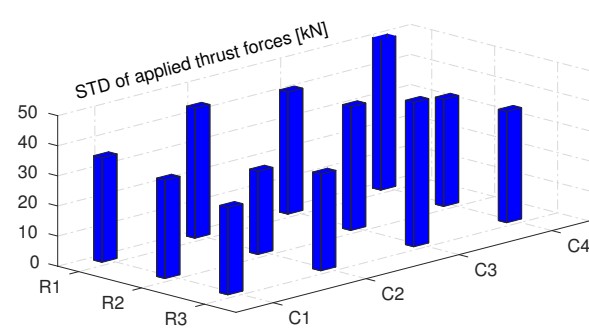

**Figure 4.** Wind farm power (left vertical axis) and deviations of the applied thrusts from their turbine average (right vertical axis) with locally greedy induction factors.

**Figure 5.** STD of the applied thrust forces on the individual wind turbines operating with the greedy induction factors $a_i = \frac{1}{3}$.

the employed ADM in the large-eddy simulation indeed reproduce lower thrust variations due to averaging the local turbulence over the rotor swept area and especially in the side rows due to wake meandering and partial wake overlaps.

**Table 2.** The key parameters of the PALM simulation set-up.

| Simulation parameter | value |
|---|---|
| Domain size $L_x \times L_y \times L_z$ | $15.3 \times 3.8 \times 1.3\,\mathrm{km}^3$ |
| Grid mesh size $N_x \times N_y \times N_z$ | $1024 \times 256 \times 128$ |
| Cell mesh resolution $\Delta_x \times \Delta_y \times \Delta_z$ | $15 \times 15 \times 10\,\mathrm{m}^3$ |
| Number of wind turbines $N_t$ | 12 |
| Wind turbine model | Actuator disc model (ADM) |
| Wind turbine control DoF | Induction factor |
| Number of grid cells per turbine | 68 |
| Atmospheric stability condition | Neutral boundary layer (NBL) |
| Ambient longitudinal wind speed $U_\infty$ at hub height | 8 m/s |
| Geostrophic wind velocity | $u = 9$ m/s and $v = -2$ m/s |
| Monin-Obukhov length scale $L$ | 3.8 km |
| Longitudinal turbulence intensity of ambient wind | $I_u \approx 5\%$ |
| Longitudinal turbulence intensity of wakes | $I_u \approx 15\%$ |
| 30-min averaged wind farm power* | 12.3 MW |
| Simulation sample time | 1 s |

\* Sum of the individual wind turbine's power production with the local greedy control setting $a_i = \frac{1}{3}$





Figure 4 summarizes the total power production and dynamic loadings of the considered case study, wherein the wind turbines operate with the locally greedy optimal control settings $a_i = \frac{1}{3}$. Note that the wind turbines always operate in the below-rated region, in which the control objective is to maximize energy capture from the incoming wind. The total turbulent wind farm power production and the corresponding time-averaged power have been used as indicators for the wind farm

available power. The thrust force which is applied on average on the wind turbines and the root mean square (RMS) of the errors across all wind turbines between the individually applied thrust forces and their turbine-averaged value are employed as indicators for uneven dynamic loadings in a waked wind farm. Figure 5 shows the standard deviations (STD) of the time-series of the applied thrusts of the individual wind turbines, operating with their locally optimal axial induction factors. Although the downwind turbines are subjected to lower wind velocities compared with the upwind turbines, they experience mainly

higher dynamic loading, which is attributed to the intensified turbulence intensity and velocity fluctuations inside the wakes. One important note is that we have focused so far on the dynamic loadings caused by wakes and their interactions with the boundary layer. In the following, we demonstrate how the wind turbine control settings, e.g., the axial induction factors, might be used to influence the structural loadings of the wind turbines in the presence of wakes with higher turbulence intensity.

## 3   Active power control design

When changing the wind farm power reference, it has to be decided how each turbine should contribute to the power production. The wake interactions among the wind turbines play a key role in the stability of the power grid and the aggregated fatigue damage of a wind farm. Therefore, two control objectives are addressed in this study. A closed-loop APC is designed to compensate wake-induced power losses and to consequently improve the quality of the wind farm power reference tracking. Depending on the amount of available power, there exist multiple solutions for APC with respect to the individual wind turbine

control inputs, i.e., the axial induction factors. These degrees of freedom (DoFs) are exploited here for coordination of the power and load distribution in order to influence the wake-induced structural loadings. At each time instance, the wind farm controller distributes a set of power demands among the individual wind turbines for realizing the overall control objectives. Figure 6 schematically shows the main components of the proposed closed-loop APC framework in this paper. A hierarchical control architecture is introduced as follows

– Level 0 stands for the open-loop wind power plant and illustrates how the employed LES model, i.e., PALM, is coupled with the $i^{\text{th}}$ wind turbine model (1)-(2) to establish the investigated wind farm case study. While a wind turbine applies the thrust force $F_a$ into the incoming flow in order to extract a certain amount of power $P$ form the wind, its wake propagates downstream and interacts with the atmospheric boundary layer.

– Level 1 contains the main components of the gain-scheduled wind turbine control (WTC) system of the $i^{\text{th}}$ wind turbine

(WT) with the main objective of locally following the power demand $P_i^{\text{dem}}$, as will be explained in subsection 3.1. Note that the wind turbine power demands are computed by the wind farm controller to cooperatively realize the control objectives at the wind farm level.



- – Level 2 represents the closed-loop APC at the wind farm (WF) level, deciding how the power productions of the individual wind turbines should be regulated in such a way that their total power production follows the power reference $P^{\text{ref}}$. In subsection 3.2, it will be demonstrated how the local power demands $P_i^{\text{dem}}$ can be adjusted in order to diminish wake-induced dynamic loadings of the wind turbines during the APC of waked wind farms.

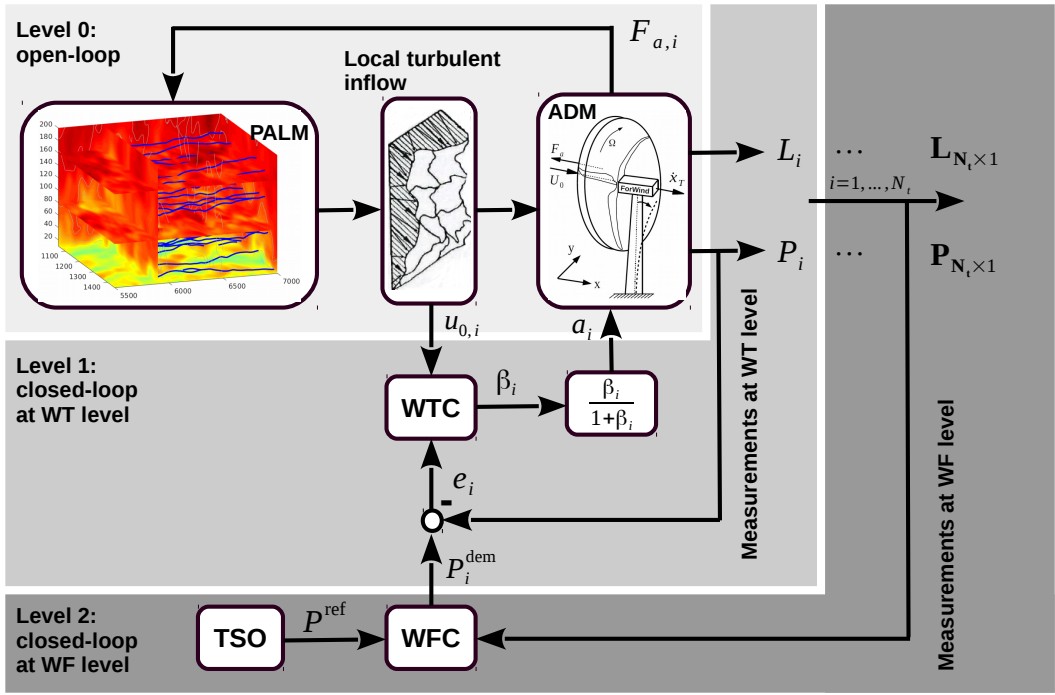

**Figure 6.** Schematic illustration of the closed-loop wind farm control (WFC) framework developed for APC of waked wind farms. The grey blocks represent the hierarchical structure of the designed APC systems at the wind turbine and the wind farm levels.

## 3.1 Wind turbine controller design

Each wind turbine has its own feedback controller such that it behaves as a dominant second-order system with a certain frequency $\omega$ and a damping ratio $\zeta$ to regulate the rate of power production (see Level 1 of Fig. 6). The control objective of the wind turbine control system is defined to locally track the power demand $P_i^{\text{dem}}$, which is commanded by the high-level wind farm controller.

The wind turbine controller is designed based on a linear representation of the aerodynamic power with respect to the control input. Hence, the following mapping is applied to the axial induction factor (Boersma et al., 2016):

$$\beta_i = \frac{a_i}{1 - a_i} \tag{9}$$





A proportional-integral (PI)-based control law is defined for APC of the $i^{\text{th}}$ wind turbine as:

$$\beta_{i,k} = K_{P,i}\; e_{i,k} + K_{I,i}\sum_{j=1}^{k} e_{i,j}, \tag{10}$$

with the local tracking error $e_{i,k} = P_{i,k}^{\text{dem}} - P_{i,k}$ at time instance $k$. Standard pole-placement is utilized for computing the gain-scheduled proportional and integral gains $K_{P,i}$ and $K_{I,i}$ as follows

$$K_{P,i} = \frac{2\tau_i \zeta \omega - 1}{\frac{1}{2}\rho A_d U_{d_i}^3}, \tag{11}$$

$$K_{I,i} = \frac{\tau_i \omega^2}{\frac{1}{2}\rho A_d U_{d_i}^3}, \tag{12}$$

with the locally rotor-averaged wind velocity $U_{d_i}$ as the scheduling variable. The closed-loop frequency and the damping ratio with the designed wind turbine controllers are set to $\omega = 0.06\,\text{Hz}$ and $\zeta = 1$, respectively. It is important to note that the gain-

scheduling approach is necessary in order to keep the closed-loop response invariant to changes in the aerodynamic control distribution. The axial induction factor is constrained to its value for maximum (greedy) power extraction, i.e., $a_i \leq \frac{1}{3}$, equivalent to $\beta_i \leq \frac{1}{2}$. Therefore, the PI-controller is extended with an anti-windup scheme for resetting the integrator component whenever the control input constraints are activated.

### 3.2 Wind farm controller design

The wind farm power reference $P^{\text{ref}}$ is distributed among the individual wind turbines on the basis of a power distribution control law, e.g., an open-loop pre-selection (Fleming et al., 2016) or a closed-loop adjustment (Vali et al., 2018c) of the power set-points. van Wingerden et al. (2017) have demonstrated that exploiting the feedback of the total wind farm tracking error improves the quality of APC and grid stability, independent of the selection of the power set-points. However, the individual wind turbine components experience different levels of the fatigue loading depending on the power set-points. In the current

study, we propose an extension to the APC of waked wind farms to actively regulate the distributing set-points, yielding reduced structural loadings when the total power production tracks a time-varying power reference, demanded by the transmission system operator (TSO). The proposed control architecture for APC with a coordinated load distribution (CLD) is depicted in Fig. 7.

### 3.2.1 Compensation of accumulated wake-induced power losses

Following van Wingerden et al. (2017), a gain-scheduled APC is designed to improve the wind farm power tracking performance by resolving undesirable local effects due to turbulence and wakes. Therefore, the control signal $\Delta P^{\text{ref}} \in \mathbb{R}$ actively adjusts the wind turbine power demands $\mathbf{P}^{\text{dem}} \in \mathbb{R}^{N_t}$ in order to compensate for the accumulated local tracking errors at each time instant $k$ as:

$$\Delta P_k^{\text{ref}} = K_P^{\text{APC-gs}}\, e_{\text{total},k}^P + K_I^{\text{APC-gs}}\sum_{j=1}^{k} e_{\text{total},j}^P, \tag{13}$$



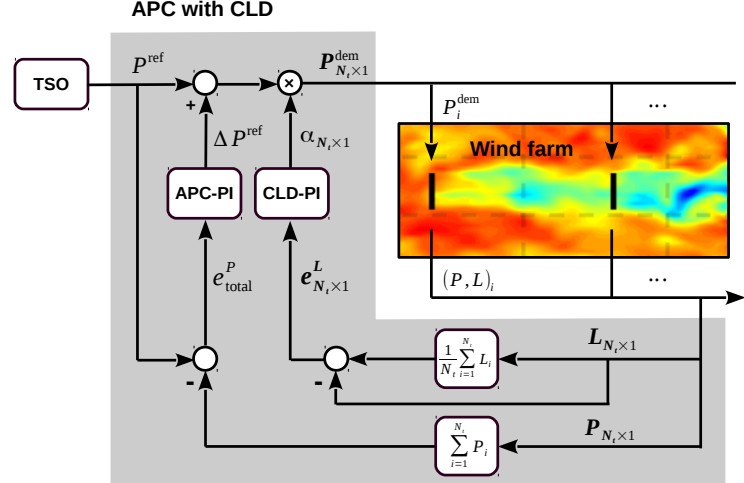

**Figure 7.** Schematic illustration of the proposed closed-loop APC of wind farms. The grey block depicts the main components of the wind farm control (WFC) block in Level 2 of Fig. 6. One feedback loop focuses on APC, while a second feedback loop provides coordinated load distribution (CLD).

with the total wind farm tracking error

$$e^P_{\text{total},k} = P^{\text{ref}}_k - \left( \sum_{i=1}^{N_t} P_{i,k} \right). \tag{14}$$

The proportional and integral gains of the APC at the wind farm level are scheduled as follows (van Wingerden et al., 2017):

$$K^{\text{APC-gs}}_{P,I} = \min \left( \frac{N_t}{N_t - N_s}, N_t \right) K^{\text{APC}}_{P,I}, \tag{15}$$

on the basis of the capabilities of the individual wind turbines in the compensation of the accumulated wake-induced power losses. Therefore, the scheduling variable $N_s$ is defined as the number of the saturated wind turbines, i.e., $a_{i,k} = \frac{1}{3}$, due to the lack of enough wind power reserve. The reader is referred to van Wingerden et al. (2017) for more details about the controller design and the closed-loop stability.

     The presented APC law (13) guarantees only a satisfactory wind farm power tracking performance when the demand remains
below the total available power of the wind farm. However, the turbulent nature of the wind and wakes might temporally cause the available wind farm power to suddenly drop below the demand, particularly for a high demand. In such condition, the individual wind turbines should indeed operate at their optimal operating point, i.e., the greedy control $a_{i,k} = \frac{1}{3}$, to locally extract the maximum amount of energy from the incoming wind. Therefore, the APC law (13) is limited to the following condition

$$\Delta P^{\text{ref}\star}_k = \left( \sum_{i=1}^{N_t} P_{i,k} \right) - P^{\text{ref}}_k \qquad \text{when} \qquad N_s = N_t. \tag{16}$$



when all the individual wind turbines are enforced to operate with the greedy induction factors, i.e., $N_s = N_t$. Indeed, the constraint (16) avoids the integrator winding up due to an abrupt drop of the total wind farm available power.

### 3.3 Wake-induced structural load control

A closed-loop power distribution law is proposed based on the structural load measurement of the individual wind turbines, the so-called coordinated load distribution (CLD). The main objective of the closed-loop APC is to minimize the total wind farm power tracking error, by e.g., adjustments of each wind turbine's power demands as follows:

$$P_{i,k}^{\text{dem}} = \alpha_{i,k} \left( P_k^{\text{ref}} + \Delta P_k^{\text{ref}} \right), \tag{17}$$

where distributing power set-points $\alpha_{i,k}$ can be chosen freely with the following constraint:

$$\sum_{i=1}^{N_t} \alpha_i = 1, \tag{18}$$

to ensure a high-quality wind farm power tracking and grid stability (van Wingerden et al., 2017; Vali et al., 2018c).

The existing DoFs are exploited here to actively adjust the power demand distribution factor $\boldsymbol{\alpha}_k \in \mathbb{R}^{N_t}$ in order to alleviate wake-induced structural loadings, while the total power production follows a wind farm power reference. Therefore, a load-based tracking error is defined for the $i^{\text{th}}$ turbine at time instant $k$ as

$$e_{i,k}^L = \left( \frac{1}{N_t} \sum_{l=1}^{N_t} L_{l,k} \right) - L_{i,k}, \tag{19}$$

describing the deviations of the instant structural loadings of the $N_t$ number of the operating wind turbines from their mean value. In the present study, we consider only the load measurements of the first tower base fore-aft bending moment, i.e., $L_i = M_i^{Ty}$, as a descriptor for the structural loadings of the $i^{\text{th}}$ wind turbine operating in a waked wind farm. Other load quantities, e.g., variation of the flap wise blade loading, short-term damage equivalent loads of the blade, main shaft response, and actuator wearing and tearing should be also considered to be more representative for a real plant. The proposed load control approach could be applied in principle as well for such more sophisticated descriptors if their measurements (or estimates) are available online.

A proportional-integral (PI)-based power distribution law is defined as

$$\alpha_{i,k} = K_P^{\text{CLD-gs}} e_{i,k}^L + K_I^{\text{CLD-gs}} \sum_{j=1}^{k} e_{i,j}^L, \tag{20}$$

for active adjustment of the power set-points, to try to level the structural loadings of the individual wind turbines. The pole-placement method is employed to design the proportional and integral gains to guarantee closed-loop stability. Within a waked wind farm, balancing the structural loading ideally yields a significant alleviation of the aggregated structural loadings, which will prolong the lifespan of the individual wind turbines. Indeed, the feedback loop (20) tends to reduce the power set-point of a highly loaded wind turbine and thus to transfer its structural loadings to other operating wind turbines.





When a power reference is relatively high compared to the wind farm available power, the downwind turbines mostly operate in high-wake conditions, leading to control saturation due to a local lack of enough wind power in reserve. Therefore, the active power control signal (13) mainly relies on the upwind turbines that can access higher wind velocities. In this situation, upwind turbines might inevitably experience higher dynamic loadings than waked turbines in order to improve wind farm

power tracking accuracy in the presence of high wake losses. Therefore, the CLD law (20) is extended with a gain-scheduling procedure as follows:

$$K_{P,I}^{\text{CLD-gs}} = \max \left( \frac{N_t - N_s}{N_t}, \frac{1}{N_t} \right) K_{P,I}^{\text{CLD}}, \qquad (21)$$

to reduce the CLD loop gain with the increase of the number of the saturated wind turbines $N_s$, indicating higher demands and strong wake conditions. In other words, the control priority is given to the APC when the size of the solution domain is limited

in order to avoid inefficient interactions of the two designed control loops. The two loops interact due to how power capture at a wind turbine affects its wake and hence the structural loadings of downwind turbines.

The proposed structural load control approach relies on the non-unique solution of the APC for wind farms. Therefore, the size of the APC solution domain plays a key role in load control. Indeed, the control solution domain is limited when a wind turbine is commanded to an unrealizable power demand, which leads to the wind turbine control saturation. In this condition,

the wind turbine structural modes may be intensively excited with high fluctuations of the wind velocities inside the wakes. In order to avoid limiting the control solution domain, the following condition is applied to the power set-point of the $i^{\text{th}}$ saturated wind turbine:

$$\alpha_{i,k}^{\star} = \frac{P_{i,k}}{P_k^{\text{ref}} + \Delta P_k^{\text{ref}}} \qquad \text{when} \qquad a_{i,k} = \frac{1}{3}, \qquad (22)$$

which sends a power demand about the locally available wind power of the saturated wind turbine. Thus, the controllability of

the problem is increased for rejecting the source of the structural loadings, e.g., the high turbulence intensity inside the wakes. One important note is that changing an operating set-point (22) also requires changes in other set-points in order to satisfy (18). Therefore, they are uniformly scaled up, when the power set-point of the $i^{\text{th}}$ saturated turbine is adjusted according to (22).

## 4   Results and discussion

This section focuses on a simulation scenario, in which the wake interactions are problematic for a good wind farm power

tracking performance, similar to Fleming et al. (2016), van Wingerden et al. (2017), and Vali et al. (2018b, c). Note that the APC of wind farms in a non-waked condition simplifies the control problem to a standard tracking one, which is not addressed here. The following three APC approaches are evaluated in this study:

–   An open-loop power distribution approach as a baseline (Baseline) (Fleming et al., 2016),

–   A closed-loop APC at the wind farm level (Ref. APC) (van Wingerden et al., 2017),

–   The proposed APC with a coordinated load distribution (APC/CLD).




A wind farm power reference tracking scenario is conducted to evaluate the APC performance of the simulated wind farm with PALM. A time-varying power reference, which is demanded from the TSO, is defined as follows

$$P_k^{\mathrm{ref}} = P_{\mathrm{farm}}^{\mathrm{base}} \left( b + c\, n_k^{\mathrm{AGC}} \right), \tag{23}$$

where the normalized perturbation of $n_k^{\mathrm{AGC}}$ is simulated using a standard test signal. Figure 8 depicts the employed normalized

RegD type of an automatic generation control (AGC) signal, the most rapidly actuating test signal which is used for APC qualification by a regional transmission organization in the eastern United States (Pilong, 2013).

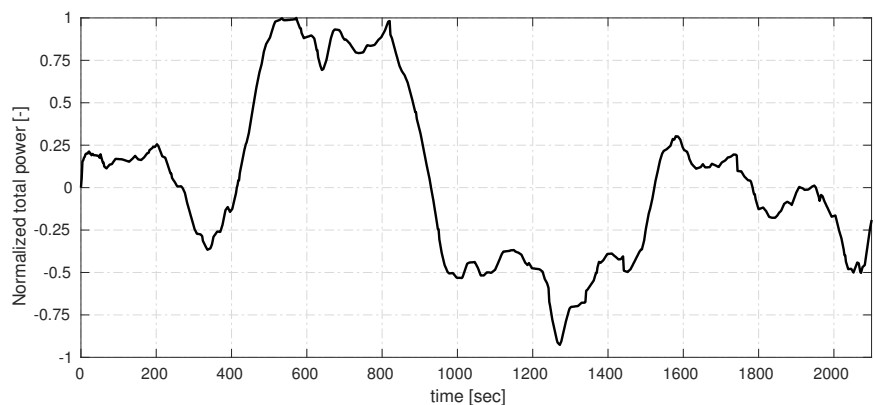

**Figure 8.** Normalized RegD type of an AGC test signal, taken from Fleming et al. (2016).

$P_{\mathrm{farm}}^{\mathrm{base}}$ is typically selected based on the time-averaged available power of a specific wind farm layout (Shapiro et al., 2017; Vali et al., 2018b). The parameters $b$ and $c$ describe the percentages of the contributions of $P_{\mathrm{farm}}^{\mathrm{base}}$ and the normalized AGC signal $n_k^{\mathrm{AGC}}$ for generating different wind farm power references. In this study, the power base is chosen as $P_{\mathrm{farm}}^{\mathrm{base}} = 12.3\,\mathrm{MW}$,

which is the time averaged power production of the simulated wind farm containing twelve wind turbines operating with the locally greedy control setting (see Fig. 4). Different levels of the power demand, which are determined using the parameter $b$, have been generated in order to investigate the performance of APC with CLD with different ratios of the power reference and the total wind power reserve. Note that the lower ratios provide a qualitative assessment of the wind farm controller at higher wind speeds, wherein the waked wind turbines have potentially more wind power reserves to follow their local demands.

However, they are still subjected to the higher structural fatigue loads due to the wake effects. In all cases, the AGC signal perturbation is considered with a maximum amplitude of $c = 10\%$ of the power base.

The analyses are focused on the AGC signal after 300 s to allow time for inflow propagation and wake development. The key parameters of the wind turbine control systems and wind farm APC with CLD are given in Table 3. The controllers are designed about the operating point of our wind farm example at an ambient wind speed of $U_\infty = 8\,\mathrm{m/s}$, with the axial induction

factor $a_i = 0.25$ for undisturbed individual wind turbines. The baseline case follows the traditional APC idea that all wind turbines are de-rated equally, i.e., $\alpha_i = \frac{1}{N_t}$. The Ref. APC exploits the feedback control signal (13) for actively regulating the





**Table 3.** The key parameters of the implemented active power controllers at the wind farm level in PALM. The wind turbine control system frequency and damping ratio are chosen as $\omega = 0.06\,\text{Hz}$ and $\zeta = 1$.

| | Pre-selected set-points | | APC-loop | | CLD-loop | |
|---|---|---|---|---|---|---|
| Baseline | $\alpha_i = \frac{1}{12}$ | [-] | NA | | NA | |
| Ref. APC | $\alpha_i = \frac{1}{12}$ | [-] | $K_P^{\text{APC}} = 3.76$ $K_I^{\text{APC}} = 0.96$ | [-] [1/s] | NA | |
| APC/CLD | NA | | $K_P^{\text{APC}} = 3.76$ $K_I^{\text{APC}} = 0.96$ | [-] [1/s] | $K_P^{\text{CLD}} = 2.80\times10^{-5}$ $K_I^{\text{CLD}} = 6.32\times10^{-7}$ | [1/Nm] [1/Nms] |

equal derating commands. The effect of the different power set-points on the power and load distribution among the individual wind turbines has been discussed in Vali et al. (2018c).

## 4.1 Active power control of waked wind farm

A central open-loop control system, which is studied in Fleming et al. (2016), is considered here as a baseline to share an AGC power signal equally among the wind turbines. Here, the wind farm tracking performance relies on APC at the wind turbine level. The quality of the baseline is sensitive to the chosen power set-points, wake interactions, local turbulence effects, and time-varying changes in atmospheric conditions. The grid stability is degraded when downstream turbines operate either fully or partially inside the wake of their upstream turbines, leading to a lack of kinetic energy that would be required for following the same fraction of the power reference. The closed-loop APC system, presented in subsection 3.2.1, is implemented in PALM coupled with the presented actuator disc models of the wind turbines. The total wind farm tracking error is fed back in order to adjust the equal derating commands against losses caused by the local wake and turbulent effects.

Figures 9 illustrates the total power productions of the simulated wind farm with the equal power set-points of $\frac{1}{12}$. The demanded power reference by the TSO is perturbed about $b = 95\%$ (upper plot) and $b = 100\%$ (lower plot) of the time averaged power production of the locally greedy control. Therefore, high wake-induced power losses of the downstream turbines adversely degrade the performance of the baseline. As demonstrated also in van Wingerden et al. (2017), the feedback controller compensates for the accumulated power losses by demanding more from turbines with higher wind power reserve. The accuracy of the APC approaches is assessed using the root mean square (RMS) of the tracking errors over the illustrated simulation run-time.

One important note is that the turbulent kinetic energy of the flow plays an important role in the closed-loop APC. For the power reference with $b = 95\%$, the closed-loop APC is able to track a power reference signal 5% larger than $P_{\text{farm}}^{\text{base}} = 12.3\,\text{MW}$ from 450 s to 850 s. However, the total power production suddenly drops due to a turbulence related strong decrease of the wind speed at about the time instant 880 s. All wind turbines operate with the locally optimal induction factor $a_i = \frac{1}{3}$ during this

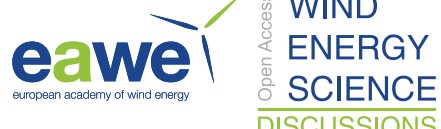



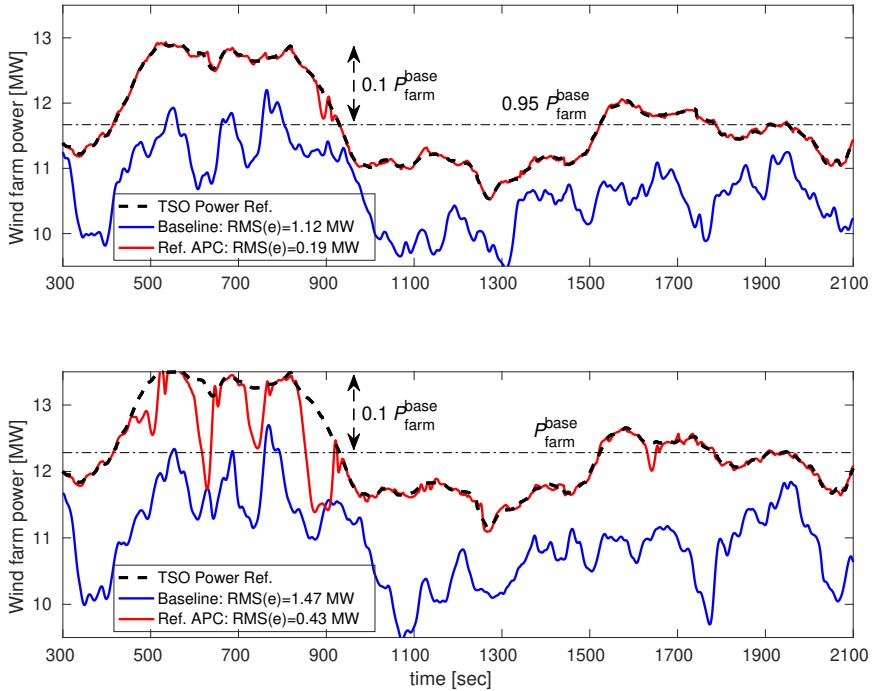

**Figure 9.** Total power production of the wind farm with closed-loop APC, compared with the baseline. Both controllers rely on constant power set-points $\alpha_i = \frac{1}{12}$, for equally distributing the AGC power reference among the wind turbines with indices of $i = 1, ..., 12$. The time-varying power reference parameters (23) are chosen here as $P_{\text{farm}}^{\text{base}} = 12.3\,\text{MW}$, $b = 95\%$ (upper plot), $b = 100\%$ (lower plot), and $c = 10\%$. $P_{\text{farm}}^{\text{base}}$ represents the time-averaged wind farm power with locally greedy induction factors $a_i = \frac{1}{3}$ (see Fig. 4).

short period of time to capture the kinetic energy from the wind as much as possible. This situation is intensified by demanding the higher power reference with $b = 100\%$, which leads to more frequently switching between the APC and the greedy control. As discussed above, the constraint (16) has been introduced to distinguish the wind farm tracking errors caused by a temporary lack of available wind farm power from a lack of power due to inefficient wake-induced losses. Indeed, the control signal $\Delta P^{\text{ref}}$

5 adjusts the TSO power reference depending on the controllability of the APC problem. A satisfactory tracking performance is achieved again as soon as the energy content of the wind increases and thus the wind farm available power goes up beyond the TSO power reference.

Figure 10 demonstrates the trajectories of the axial induction factors of the individual wind turbines for following the power reference with $b = 95\%$ in the most critical time span up to 1300 s. The closed-loop APC mostly saturates the downwind

10 turbines to the locally optimal operating point $a_i = \frac{1}{3}$, which means that the controllability of the problem is limited for high power references. Thus, they are not responsive enough for rejecting the sources of dynamic loadings, e.g., a high turbulence intensity inside the wake, through their control inputs. High wake losses are mainly compensated by the upwind turbines




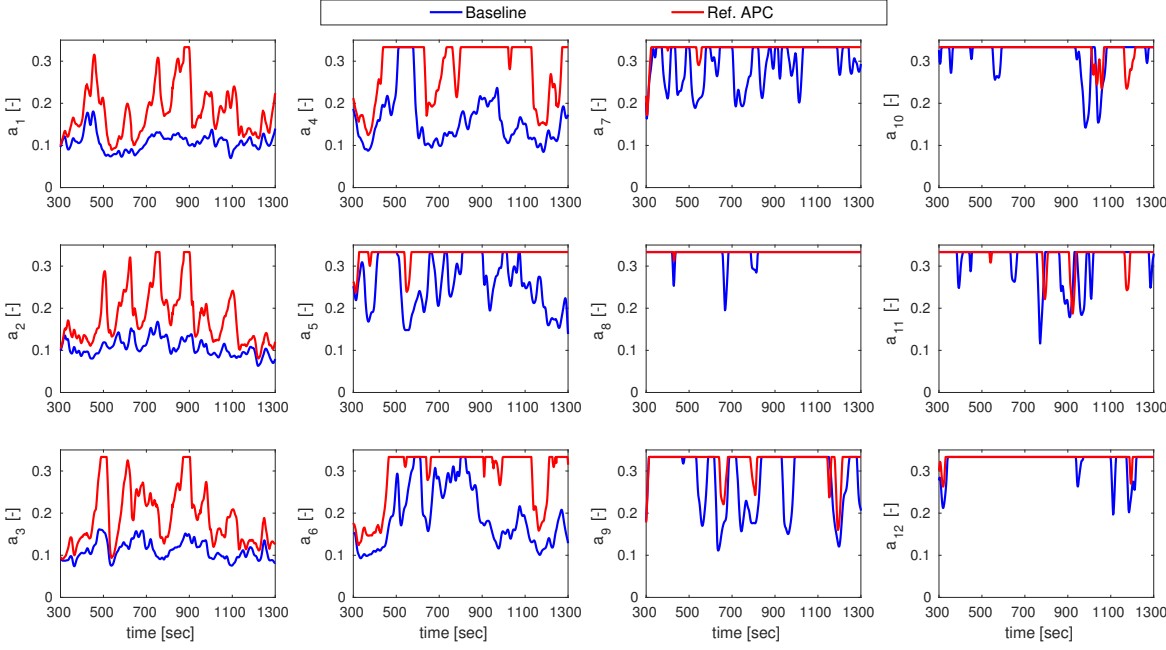

**Figure 10.** The axial induction factor trajectories of the individual wind turbines with the baseline and closed-loop APC for the power reference with $P_{\text{farm}}^{\text{base}} = 12.3$ MW, $b = 95\%$, and $c = 10\%$. Both controllers rely on equal power set-points, i.e., $\alpha_i = \frac{1}{12}$.

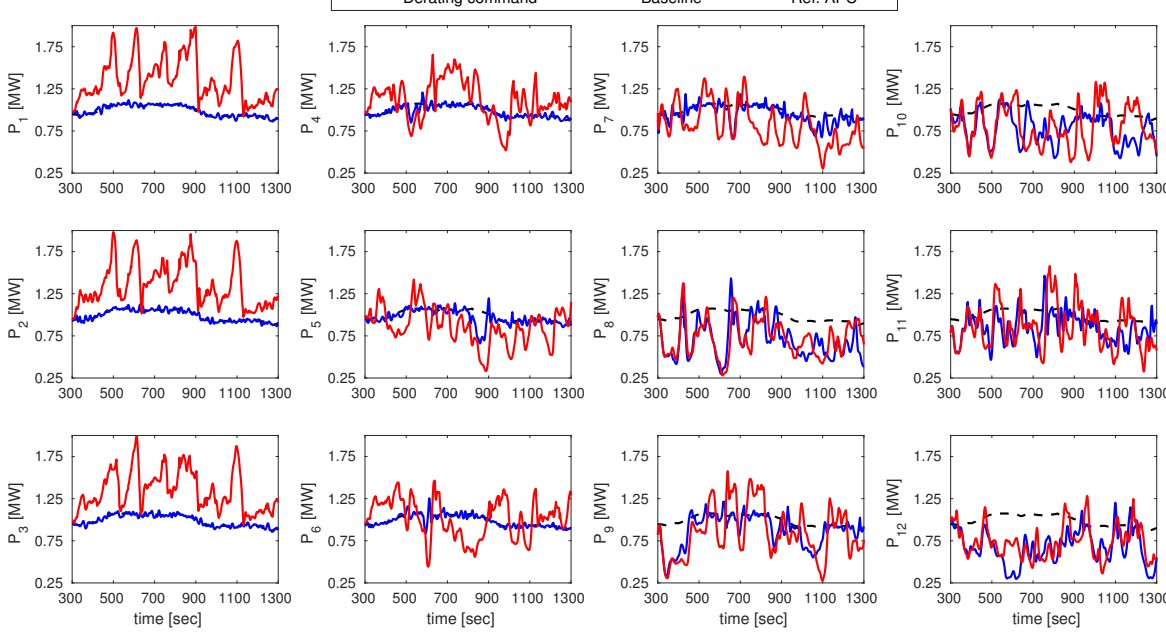

**Figure 11.** The power productions of the individual wind turbines with the baseline and closed-loop APC for the power reference with $P_{\text{farm}}^{\text{base}} = 12.3$ MW, $b = 95\%$, and $c = 10\%$. Both controllers rely on equal power set-points, i.e., $\alpha_i = \frac{1}{12}$.





(first column) using the total wind farm power feedback. Indeed, high structural loadings are expected on the upwind turbines because of their large power variations to compensate for power losses at the waked wind turbines.

Figure 11 plots the corresponding power productions of the individual wind turbines. Both wind farm controllers apply the concept of equal power derating (dashed black curves), which is then adjusted in the closed-loop APC. The upwind turbines
1, 2, and 3 mainly contribute to the wind farm power tracking performance, as shown in Fig. 9. Moreover, the second row of the wind turbines, containing wind turbines 2, 5, 8, and 11 (see Fig. 3), interact fully with their wakes. Compared with the baseline, higher losses of the 5th turbine are caused by the higher energy extraction of the 2nd turbine. However, the flow is recovered faster with the closed-loop APC when it reaches the 8th wind turbine, which operates at the greedy operating point for both cases (see Fig. 10).

## 4.2  Active power control with coordinated load distribution law

Applying our new control law, the distributed power set-points are actively adjusted at each time instant to achieve smaller deviations of structural loading of the individual wind turbines from their mean value. In the previous section, it has been discussed that a demand at $b = 95\%$ of $P_{\text{farm}}^{\text{base}} = 12.3$ MW limits the flexibility and the APC solution domain relative to a lower demand $b$. Therefore, we elaborate the performance of APC with CLD with different parameterizations of the power reference (23) in
order to enlarge the domain of APC solution.

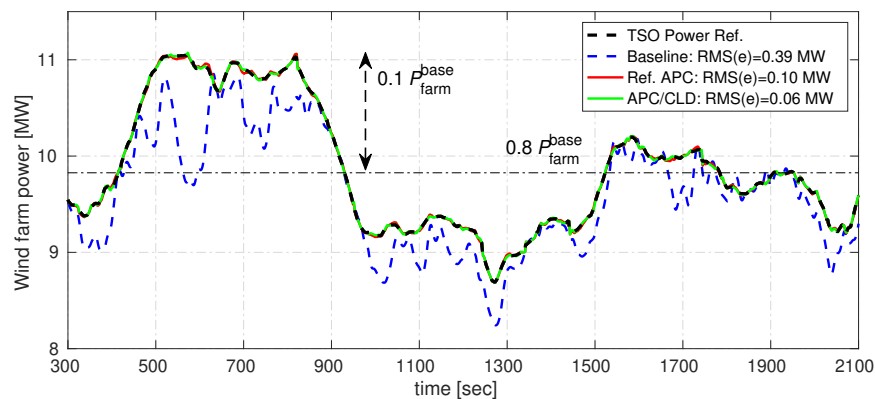

**Figure 12.** Total power production of the wind farm with closed-loop APC with coordinated load distribution (CLD). The time-varying power reference parameters (23) are chosen here as $P_{\text{farm}}^{\text{base}} = 12.3$ MW, $b = 80\%$, and $c = 10\%$. $P_{\text{farm}}^{\text{base}}$ represents the time-averaged wind farm power with locally greedy induction factors $a_i = \frac{1}{3}$ (see Fig. 4).

Figure 12 plots the AGC response and indicates the RMS of the wind farm power tracking error from 300 s to 2100 s for the studied APC approaches, where the demanded power reference from the TSO is perturbed about $b = 80\%$ of $P_{\text{farm}}^{\text{base}} = 12.3$ MW with $c = 10\%$ of the normalized AGC signal. The tracking error of the baseline (dashed blue curve) is reduced due to the lower wake deficits. However, the wake-induced power losses still degrade the grid stability and this can be addressed using the





closed-loop APC (solid red curve). Note that the power set-points are chosen as $\frac{1}{12}$ as before for both cases. The wind farm power tracking accuracy is further improved when the power set-points are actively adjusted using the proposed CLD control law (green dashed curve).

Figure 13 depicts the trajectories of the regulated power set-points. The APC with CLD exploits the flexibility of the wind
turbine control inputs in order to find the APC solution that yields a better balance of structural loadings for the individual wind turbines. Contrary to the equal set-points of $\frac{1}{12}$, the power set-points of the upstream turbines 1 to 3 are significantly increased because they are operating in the free stream with high power reserves and low turbulence intensity (of approximately 5%). Therefore, there exists the possibility to increase their contributions toward structural load reduction of the downwind turbines operating inside wakes with higher turbulence intensity (of approximately 15%). Furthermore, the set-points of the saturated
wind turbines are adjusted according to (22), which causes the wind farm controller to demand these waked wind turbines to produce about their locally available wind powers.

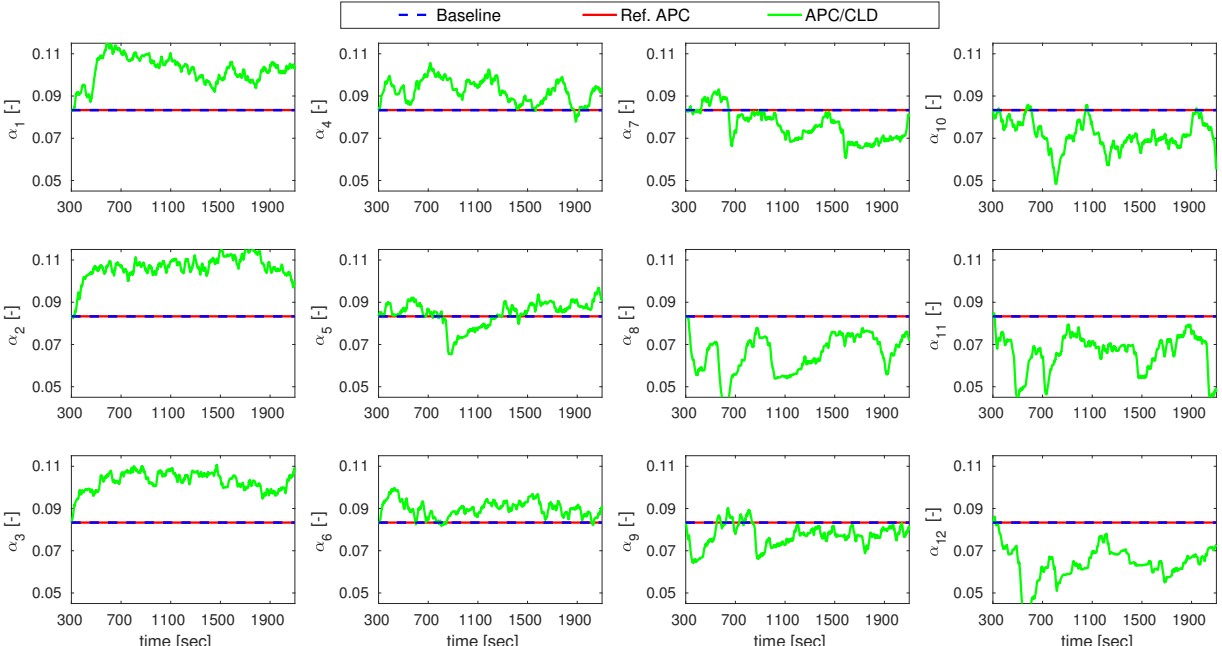

**Figure 13.** The trajectories of the power set-points with APC with coordinated load distribution (CLD). The baseline and Ref. APC rely on equal power set-points, i.e., $\alpha_i = \frac{1}{12}$.

Figures 14 and 15 reveal the impact of the CLD law on the wind farm power tracking and the dynamic loading, respectively. The first figure illustrates the control signal $\Delta P^{\mathrm{ref}}$, which actively regulates the wind turbine power demands in order to compensate for the total wake-induced power losses. Contrary to the reference case (red curve), APC with CLD demands
significantly smaller corrections via the overall power demand, and the power set-points are adjusted more using the local power and structural load measurements of the individual wind turbines. This addresses the accuracy improvement of the wind farm power tracking, compared with the reference closed-loop APC. Note that the APC and CLD loops are activated at



time instants $200\,\mathrm{s}$ and $300\,\mathrm{s}$, respectively. Each distributing signal $\alpha_{i,k}$ is initialized with an equal power set-point of $\frac{1}{12}$, as illustrated in Fig. 13.

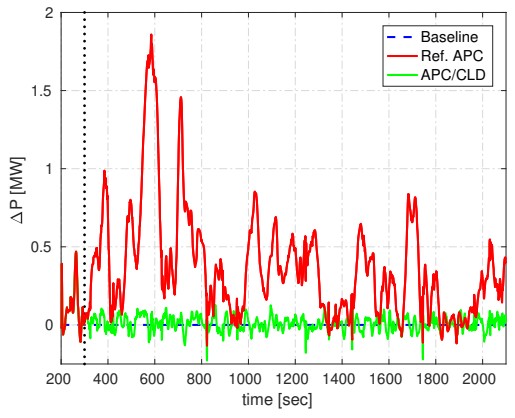

**Figure 14.** The control signal $\Delta P^{\mathrm{ref}}$ for adjustment of the wind turbine power demands.

**Figure 15.** RMS errors between the applied thrust of the individual wind turbines and their mean value.

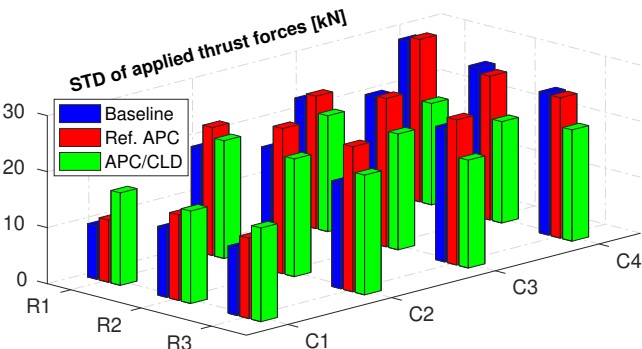

**Figure 16.** The standard deviation (STD) of the applied thrust forces on the individual wind turbines for the different APC approaches. R and C stand for the row and column of wind turbines, respectively, for our simulated wind farm (see Fig. 3). The power reference (23) is parameterized as $P_{\mathrm{farm}}^{\mathrm{base}} = 12.3\,\mathrm{MW}$, $b = 80\%$, and $c = 10\%$. The Ref. APC without CLD (red bars) and the APC with CLD (green bars) lead to almost the same accuracy of the wind farm power tracking (cf. Fig. 12).

Figure 15 shows the pattern of dynamic loadings of the individual wind turbines for the studied APC approaches. At each time instant, the root mean square (RMS) of the errors across all wind turbines between the applied thrust forces and their mean value is calculated. The standard deviation (STD) of the applied thrust forces on all wind turbines from $300\,\mathrm{s}$ to $2100\,\mathrm{s}$ is also denoted in the legend for each approach. Although the reference APC improves the accuracy of the tracking performance by almost 74% (see Fig. 12), the loading pattern remains similar to the baseline (see solid red and dashed blue curves in Fig. 15)



due to the same selection of the power set-points. The reader is referred to Vali et al. (2018c) for analyses on the dependency of the thrust distribution patterns on different stationary selections of the power set-points. The increase of the STD of the exerted thrust forces is related to the compensation of wake-induced power losses using feedback. As demonstrated, APC with CLD is capable of regulating the wind turbine power productions with at least the same quality of AGC response of the Ref.

APC without CLD. The RMS of the defined thrust-based error over time (see green line in Fig. 15) and the associated STD of the exerted thrust forces (see the legend in Fig. 15) are significantly reduced, meaning that the wind turbines are loaded more evenly.

Figure 16 compares the STD of the applied thrust forces on the individual wind turbines from 300 s to 2100 s. APC with CLD (green bars) results in a reduction on deviations of the thrust forces of the downwind turbines operating inside the wakes,

compared with the baseline and Ref. APC. In total, the aggregated structural loading is remarkably lowered by slight load transfer to the upwind turbines, i.e., the first column (C1). Such an increase of the dynamic loading of these turbines is not critical since they are operating in a low ambient turbulence anyway.

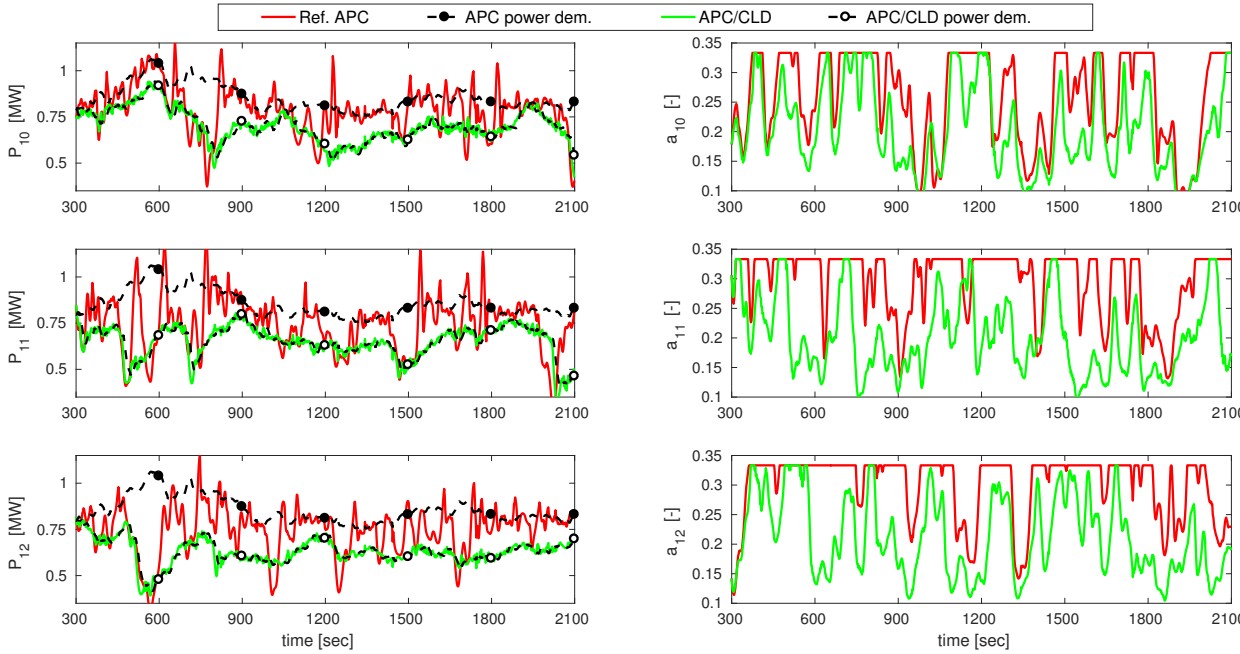

**Figure 17.** The adjusted power demands (left) and the corresponding trajectories of the axial induction factors (right) of the last column (C4) of turbines in the wind farm. Both APC approaches provide almost the same quality of AGC response (see Fig. 12). The power reference (23) is parameterized as $P_{farm}^{base} = 12.3$ MW, $b = 80\%$, and $c = 10\%$.

Figure 17 provides a closer view on the trend of the proposed CLD law for the reduction of wake-induced structural loads, by e.g., looking at the performance of the wind turbines 10 to 12 in the last wind farm column, compared with the Ref. APC

without CLD. The adjusted power demands with the APC law (13) are actively derated over time using feedback from the





local power and structural load measurements. It should be noted that the wind farm power tracking performance is kept unchanged through uprating the wind turbines which are subjected to lower external turbulent excitations, particularly the upwind turbines (see Fig. 13). Indeed, limiting the power demands increases the controllability of the waked wind turbines. As shown, the axial induction factors are saturated less often and hence have more freedom (see green curves) to reject

the intensified fluctuations due to the turbulence, which is the main source of the wake-induced structural loadings on the downstream turbines.

### 4.3   Performance analyses

This section evaluates the applicability of the proposed APC with CLD from some practical criteria, e.g., the APC accuracy, the extreme loading, and the aggregated fatigue loading. The wind turbine structural fatigue loading is among the key factors

in the design of wind turbines (IEC, 2005). The fatigue load analysis is achieved by comparing the load spectra obtained from Rainflow-counting of the time series of the stresses with a characteristic curve of the design resistance of the component under investigation, the so-called S–N or Wöhler curve. This study only provides a qualitative assessment of the corresponding damage equivalent load (DEL) of the tower base fore-aft bending moment as a descriptor for structural fatigue loading in wind farms. As shown in section 2.2, it is straightforward to extend the ADM in the PALM simulation code with the tower structural

dynamics (2). Other fatigue load sources, e.g., quasi-periodic load disturbances due to wakes partially overlapping the rotor swept area, will be more representative for a comprehensive fatigue load analysis when they are included inside the simulation model. The corresponding DELs are computed using the 30-minute time series (from 300 s to 2100 s) of the tower base fore-aft bending moment. The inverse slope of the S–N curve is considered as $m = 4$, which is commonly used for steel components like the tower.

Table 4 summarizes the performance results of the studied APC approaches. Each case is evaluated with four AGC-based power references (23) with different parameterizations of $b$ in order to investigate the effect of wind power reserves and thus different sizes of the APC solution domains. The accuracy of APC is evaluated first with the root mean square (RMS) of the wind farm power tracking error. Then, the standard deviations (STD) of the applied thrust forces on all wind turbines are outlined as well. The possible impacts on the structural loadings of the wind farm are analyzed using the maximum amplitude

and the corresponding DELs of the fore-aft tower base bending moment of all twelve wind turbines, which are plotted in Fig. 18 for all case studies. We introduce the mean DEL and the standard deviation (STD) of the corresponding DEL as two indicators for analyzing the wind farm fatigue loading as a whole and the variation of fatigue load distributions among the wind turbines, respectively. The local quantity is of special interest since particular turbine situations with higher DELs are not balanced out by other periods with lower DELs due to the strong non-linearity of the fatigue damage on the stress amplitude.

Figure 19 and 20 provide a comparative insight into the overall performances for achieving the defined control objectives at the wind farm level. The root mean square (RMS) of the wind farm power tracking error (Fig. 19), the mean DEL (left plot of Fig. 20), and the standard deviation (right plot of Fig. 20) of the corresponding DELs are normalized with respect to the baseline case with the power reference level $b = 80\%$. The bar graphs reveal the relative performance of the studied APC approaches with different levels of the TSO power reference and the corresponding wake effects. In general, higher





**Table 4.** Performance assessment of the studied APC approaches with PALM for different power reference parameterizations. The time-varying power references (23) are parameterized as $P_{\text{farm}}^{\text{base}} = 12.3\,\text{MW}$, $b = 80\%$, $85\%$, $90\%$, $95\%$, respectively, and $c = 10\%$. All quantities are computed for the 30-minute time series (from $300\,\text{s}$ to $2100\,\text{s}$) of the wind farm operation.

| | | RMS of total power tracking error [MW] | STD of thrust force [kN]* | Max of tower base moment [MNm]* | Mean DEL of tower base moment [MNm]* | STD of DEL of tower base moment [MNm]* |
|---|---|---|---|---|---|---|
| Baseline | $b = 80\%$ | 0.390 | 31.623 | 38.576 | 2.626 | 1.160 |
| | $b = 85\%$ | 0.556 | 32.330 | 37.993 | 2.798 | 1.098 |
| | $b = 90\%$ | 0.784 | 33.540 | 37.638 | 3.015 | 1.133 |
| | $b = 95\%$ | 1.120 | 34.113 | 38.614 | 3.176 | 1.131 |
| Ref. APC | $b = 80\%$ | 0.104 | 32.803 | 37.349 | 2.841 | 0.985 |
| | $b = 85\%$ | 0.106 | 34.476 | 38.830 | 3.249 | 1.037 |
| | $b = 90\%$ | 0.150 | 36.288 | 44.057 | 4.152 | 0.759 |
| | $b = 95\%$ | 0.192 | 41.857 | 47.243 | 4.689 | 0.388 |
| APC with CLD | $b = 80\%$ | 0.066 | 20.378 | 30.539 | 2.152 | 0.388 |
| | $b = 85\%$ | 0.074 | 24.738 | 31.632 | 2.590 | 0.389 |
| | $b = 90\%$ | 0.084 | 34.117 | 41.087 | 3.263 | 0.263 |
| | $b = 95\%$ | 0.128 | 52.543 | 50.512 | 4.172 | 0.552 |

* The quantity is calculated using the applied thrust forces or tower base load measurements of all wind turbines.

power demands create stronger wake effects downstream, e.g., larger wind velocity deficits and fluctuations, which lead to accumulated power tracking errors and higher structural fatigue loadings, respectively.

### 4.3.1 Baseline open-loop control

The baseline case, i.e., an open-loop power distribution law, clarifies the wake challenge of the APC problem when the TSO power demand increases (see the blue bars of Fig. 19). A higher level of the power reference $b$ obviously demands more energy extraction from all turbines, which cannot be realized by the waked ones with lower kinetic energy content and undesirably increases total wind farm tracking errors. In addition, higher local demands and wake effects slightly increase the mean DEL. When a downwind turbine operates inside a wake with lower energy content than local demand, it is limited to its maximum operating point, i.e., the greedy control setting $a = \frac{1}{3}$. In such condition, there exists no control freedom for reducing wake-induced structural loadings. Higher DELs for the last two columns of turbines (C3) and (C4) correspond to their operation inside strong wakes with limited controllability (see blue bars of Fig. 18). Note that the baseline case commands all wind turbines equally. Therefore, the first column of turbines (C1) experiences the lowest dynamic loadings because these turbines operate in free-stream with lower turbulence intensity and higher power reserve for local power tracking. The wind turbines of

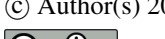



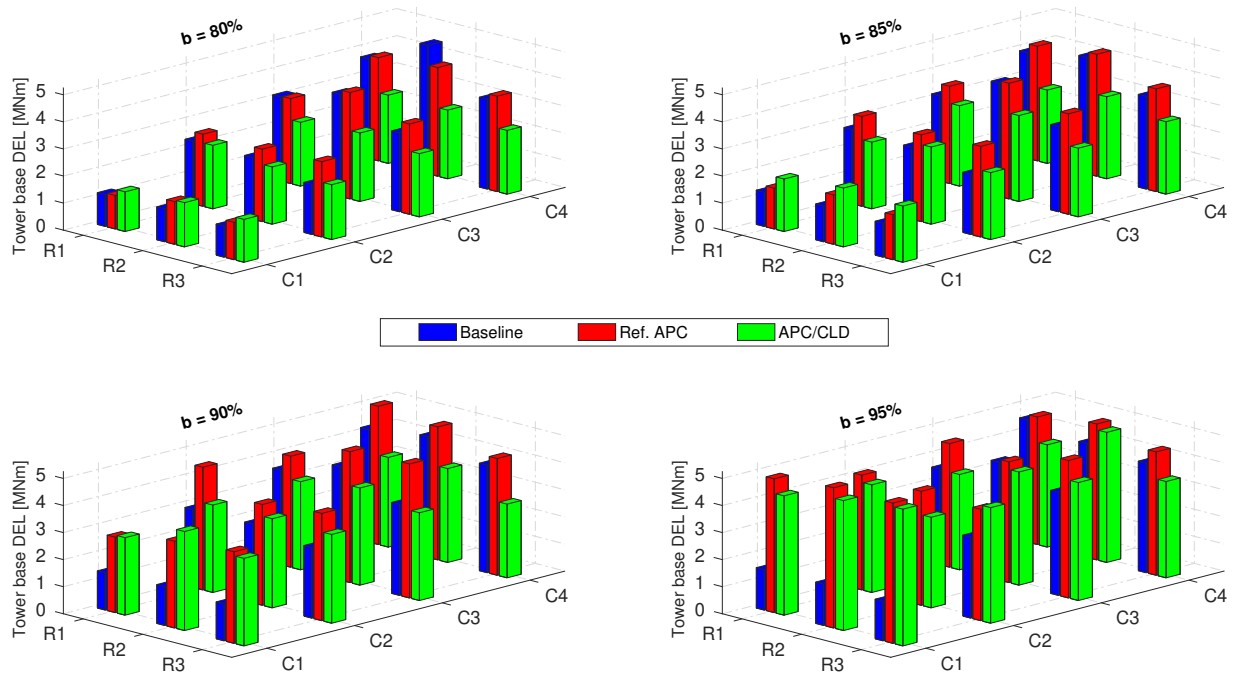

**Figure 18.** Corresponding DELs of the tower base fore-aft bending moment with the APC approaches for different levels of AGC power demand (23), parameterized as $P_{\mathrm{farm}}^{\mathrm{base}} = 12.3$ MW, $b = 80\%, 85\%, 90\%, 95\%$, respectively, and $c = 10\%$. R and C are referring to the turbine positions inside the simulated wind farm (see Fig. 3).

the second column (C2) are subjected to wakes with higher kinetic energy than their local demands, yielding medium wake-induced fatigue loadings. Thus, the baseline APC represents an uneven distribution of structural loadings at different columns of wind turbines, as it leads to the highest standard deviations of the corresponding DELs, compared with the closed-loop APC approaches (see right plot of Fig. 20).

### 4.3.2 Closed-loop APC

The closed-loop APC is able to compensate the accumulated power tracking errors due to wake losses, in return for higher dynamic loadings of the wind farm on average (see the red bars). It is discussed above that large power variations are demanded from upwind turbines to improve the total power tracking performance in the high-waking condition when $b = 95\%$ (see Fig. 11); the increased DELs of the upwind turbines (see red bars of Fig. 18) are due to their contributions when using feedback at the wind farm level. As illustrated in the right plot of Fig. 20 (see red bars), the load distribution patterns remain close to the baseline for lower demands, e.g., $b \leq 85\%$, due to higher wind power in reserve. However, the wind turbines are loaded with a higher mean DEL but more even distribution when the level of power demand increases, e.g., $b \geq 90\%$. Indeed, such higher

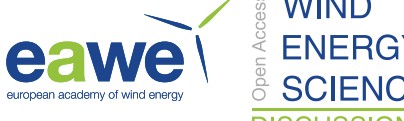

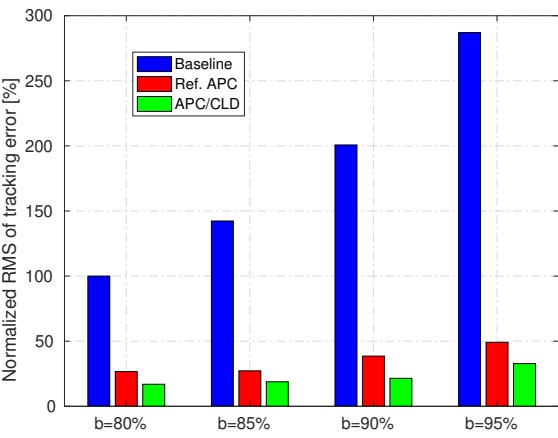

**Figure 19.** Normalized root mean square (RMS) of wind farm power tracking error for different levels of AGC power reference (23), parameterized as $P_{\text{farm}}^{\text{base}} = 12.3$ MW, $b = 80\%, 85\%, 90\%, 95\%$, respectively, and $c = 10\%$. The baseline case with power demand level $b = 80\%$ is considered as the reference.

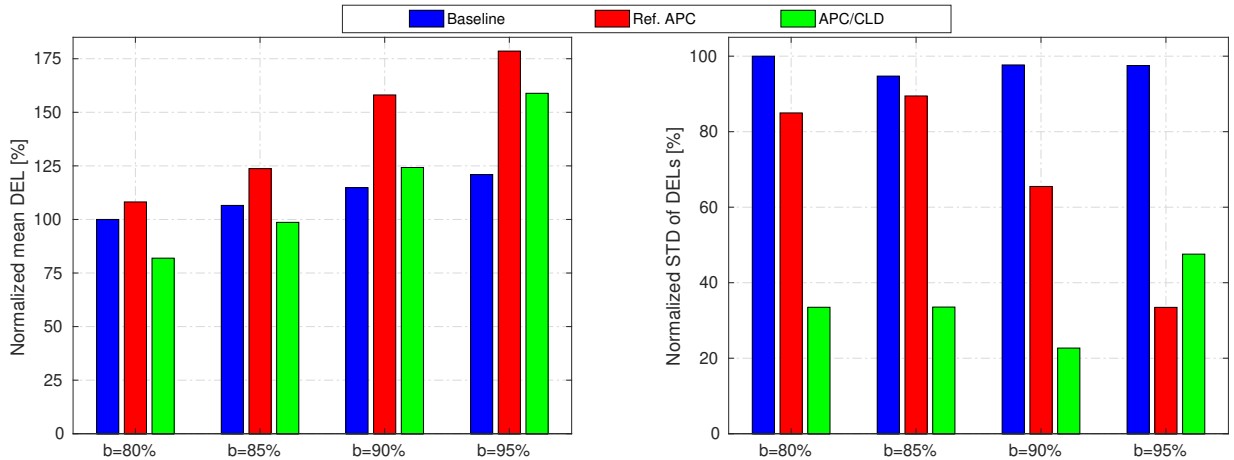

**Figure 20.** Normalized mean (left) and standard deviation (right) of the corresponding DELs of the tower base fore-aft bending moment for different levels of AGC power reference (23), parameterized as $P_{\text{farm}}^{\text{base}} = 12.3$ MW, $b = 80\%, 85\%, 90\%, 95\%$, respectively, and $c = 10\%$. The baseline case with power demand level $b = 80\%$ is considered as the reference.

demands limit the controllability of the APC problem to the upwind turbines, yielding significant load transfers closer to those of strongly waked turbines (see Fig. 18).



### 4.3.3 APC with coordinated load distribution

Extending the APC with the proposed coordinated load distribution (CLD) law improves both the tracking performance and the reduction of structural loadings of the waked wind farm (see the green bars). The APC with CLD exploits the redundancy of the control problem to search for the APC solution that alleviates wake-induced structural loadings of the wind turbines.

Therefore, the domain of APC solution plays a significant role in a satisfactory performance of the proposed APC with CLD. Three main features of the proposed CLD loop for an effective load coordination over the whole range of the TSO power demands are:

– Levelizing dynamic loadings on the individual wind turbines through (20) when there exists enough wind power in reserve for tracking the TSO power reference,

– Reducing the CLD loop gain through the gain-scheduling procedure (21) when upwind turbines are highly loaded as they compensate for the accumulated wake-induced tracking errors, and

– Increasing the controllability of the saturated wind turbines through (22) by adjusting the power set-points according to locally available wind power at the waked wind turbines.

Lower power demands, e.g., $b = 80\%$ and $b = 85\%$, benefit from higher wind power reserves, which enlarge the APC

solution domain and thus yield higher chances for fair balances of structural loadings and reducing the mean DEL (see left plot of Fig. 20), even lower than the baseline. The reduced STDs of the corresponding DELs verify that the APC with CLD leads to more even fatigue load distributions among the wind turbines (see right plot of Fig. 20), compared with the other APC approaches. This feature might be an efficient concept for operating in the above-rated region, wherein the wind power reserves are high enough for a high-quality tracking performance. One important note is that the fatigue load distribution

pattern is levelized more evenly for the demand case $b = 90\%$, while its corresponding mean DEL is increased. Indeed, larger power variations of the upwind turbines cause their fatigue loadings to be closer to those of the strongly waked downwind turbines (see Fig. 18 for more details).

A further aspect to be discussed is that a very high power demand, e.g., $b = 95\%$, limits the APC solution domain to the upwind turbines, as shown in Fig. 10. Almost nine out of the twelve wind turbines, i.e., all downstream turbines, are saturated

during most of the simulation due to lack of enough wind power, and this yields highly increased dynamic loading of the upwind turbines, even more than waked ones (see Fig. 18). In such conditions, the CLD loop ineffectively sacrifices the accuracy of the power reference tracking. Thus, the gain-scheduling (21) is used to reduce the CLD loop gain and to avoid inefficient interactions of the two control loops. As shown in Fig. 20, the APC with CLD reduces the mean DEL while the STD of the corresponding DELs increases, compared with the reference APC. In this case, the structural load mitigations

have been locally achieved through adjustments of the power set-points in accordance with the available wind power inside the wakes. The increased control solution domain provides chances to react against the intensified wind velocity fluctuations inside the wakes. As discussed above, adjusting the power set-points of the nine waked turbines according to their available powers demands an increase on the power set-points of the three upwind turbines to satisfy (18) for a reliable APC. That is



why the STD of the applied thrust forces on all wind turbines and the maximum amplitude of the resultant first fore-aft tower
base moments are increased compared with the Ref. APC.

## 5   Conclusions

A new APC approach has been introduced and successfully demonstrated to reduce the wake-induced structural loads of a
waked wind farm while the sum of their actual power productions tracks a time-varying wind farm power reference. Since there
exist multiple solutions for the APC problem with respect to the distribution of the individual wind turbine power demands, it
is obvious that a solution with mitigated structural loading should be possible. The employed distribution law actively adjusts
the individual wind turbine power set-points using feedback from the local power and structural load measurements. The study
highlights that the enlargement of the APC solution domain increases the controllability of the system for rejecting the sources
of the intensified dynamic loadings inside wakes. The accuracy of the wind farm power tracking is further improved when the
power set-points are chosen to enlarge the APC solution domain. When a wind turbine is commanded with a realizable power
demand, instead of being saturated with respect to the induction factor, i.e., $a_i = \frac{1}{3}$, its controllability for rejecting wake-
induced structural loadings is increased as well. A gain-scheduling approach is introduced to avoid inefficient competition
between the two designed control loops when the overall power demand is high.

Simulation results show that a high-quality AGC response and a more even structural load distribution can be achieved
simultaneously when there exists sufficient wind power in reserve. Demanding higher power production from the TSO limits
the flexibility of the APC problem with respect to the wind turbine control inputs due to their saturation. Thus, a satisfactory
wind farm power tracking performance may require higher dynamic loadings on the upwind turbines because of tremendously
accumulated wake-induced errors. However, the mean fatigue loading of a wind farm can still be reduced by effectively
adjusting the local power demands at each wind turbine. In the future, the proposed active power control approach will be
examined from more reliable and practical perspectives, e.g., fatigue load analysis, using large-eddy simulations coupled with
aeroelastic wind turbine models. This is of particular interest with respect to asymmetric rotor loads induced by partial wake
overlap, which was beyond the scope of the present analysis.



*Acknowledgements.* Thanks are given to Róbert Ungurán for his kind support on establishing the communication between MATLAB and the PALM code.

5    This work has been partly funded by the Federal Ministry for Economic Affairs and Energy according to a resolution by the German Federal Parliament ("WIMS-Cluster" 0324005) and by the Ministry for Science and Culture of Lower Saxony through the funding initiative "Niedersächsisches Vorab" (project "ventus efficiens"). Support from the Hanse-Wissenschaftskolleg in Delmenhorst, Germany and from a Palmer Endowed chair at the University of Colorado Boulder is also gratefully acknowledged.



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
