# Peer review of "An active power control approach for wake-induced load alleviation in a fully developed wind farm boundary layer"

_Wind Energy Science, 2018_

## Short Comment (SC1) · 6 Dec 2018

Dear authors,

I was pleased to see this work and I believe that the authors are working on an important and interesting control problem. However, I also believe that a significant amount of literature is missing. For example:

1) V. Spudic, C. Conte, M. Baotic, M. Morari, "Cooperative distributed model predictive control for wind farms", Optimal Control Applications and Methods, 2014.

2) S. Siniscalchi-Minna, F. D. Bianchi, M. De Prada Gil, C. Ocampo-Martinez, "A wind

farm control strategy for power reserve maximization", Renewable Energy, 2018.

3) H. Zhao, Q. Wu, Q. Guo, H. Sun, Y. Xue, "Distributed model predictive control of a wind farm for optimal active power controlpart I: Clusteringbased wind turbine model linearization", IEEE Transactions on Sustainable Energy, 2015.

4) T.N. Jensen, T. Knudsen, T. Bak, "Fatigue minimising power reference control of a de-rated wind farm" Journal of Physics: Conference Series, 2016.

5) A.D. Hansen, P. Sørensen, F. Iov, and F. Blaabjerg, "Centralised power control of wind farm with doubly fed induction generators", Renewable Energy, 2006.

6) B. Biegel, D.D. Madjidian, V. Spudic, A. Rantzer, J. Stoustrup, "Distributed low-complexity controller for wind power plant in derated operation", International Conference on Control Applications, 2013.

7) D. van der Hoek, S. Kanev, W. Engels, "Comparison of Down-Regulation Strategies for Wind Farm Control and their Effects on Fatigue Loads", American Conrtol Conference, 2018.

8) S. Siniscalchi-Minna, F. D. Bianchi, C. Ocampo-Martinez, "Predictive control of wind farms based on lexicographic minimizers for power reserve maximization", American Control Conference, 2018.

9) S. Boersma, B.M. Doekemeijer, S. Siniscalchi-Minna, J.W. van Wingerden, "A constrained wind farm controller providing secondary frequency regulation: an LES study", Renewable Energy, 2018.

These are all articles that deal with wind farm APC/reference tracking/derate strategies, though are all not addressed. The last article in the list is even using the same high-fidelity code as the submitted work.

I believe that the above literature should be addressed in the submitted manuscript.

Regards

---

## Referee Comment (RC1) · Anonymous Referee #1 · 20 Dec 2018

The paper proposes a new method for providing AGC from a wind farm while accounting for wake interactions that increases the objectives of the wind farm from power tracking to include minimization of structural loads. Structural loading is limited to be only tower fore-aft, but this is a reasonable decision for this stage of research.

Overall the paper quality is very high, the text is easy to follow and explanations are thorough. The figures are clear, and the captions provide good detail. Tables and formulas make clear the methods and parameters. Abstract and introduction flow well. Finally, the reference to, and continuous interaction with the existing literature is truly excellent, and the paper very clearly indicates the state of the literature, and what

contributions it is making.

I therefore find the paper close to ready as is. One thought I had reading the paper is that perhaps the problem's complexity will grow in a new dimension as additional loads are added, as it is possible that control decisions which help one load, hurt another, and therefore the relative importance weighting between loads will add some complexity (perhaps a necessary arbitrary selection will need to be made, or perhaps some sort of pareto analysis could stand in). Still, this is clearly and properly identified as future work, and so I only comment on it here in case the author's have already thought about this.

Minor comments:

1) The paper is somewhat long, if it is possible to find opportunities to trim which don't detract too severely from the overall quality, it could be worthwhile

2) The line on page 10 "The wake interactions among the wind turbines play a key role in the stability of the power grid" struck me while reading it as perhaps over-stating things.
* * *

---

## Referee Comment (RC2) · Anonymous Referee #2 · 2 Jan 2019

The paper is difficult to follow. Nomenclature is not obvious, especially all the subscripts/superscripts. Simple verbal explanations of equations would help. I miss a good concise summary of the specific questions the paper is trying to answer, and the specific ways in which the paper advances the topic. The paper points out that only tower loads are looked at, which is reasonable as a starting point, but how accurately is this assessed? The paper gives ambient turbulence intensity of $\sim$5% and waked turbulence intensity of $\sim$15%, but wake turbulence is inhomogeneous, and this does not say anything about the way turbulence varies across and along the wake, or in multiple wakes, nor about the frequency content or length scales of the additional turbulence, all of which will affect tower loads. Does PALM actually deal with all this complexity

properly, so that the ∼5% and ∼15% are just given for interest? Equation (10): How does beta relate to the pitch and torque demands? A given beta can be achieved with different combinations of pitch and torque, and this affects loads. How does this relate to a practical wind turbine controller? End of section 3.2.1: "In such condition, the individual wind turbines should indeed operate at their optimal operating point, i.e., the greedy control": does this ignore the possible (admittedly disputed) potential benefits of wake induction control?

Editors: because the paper is hard to follow, I don't have time to complete the review as thoroughly as I'd like.

---

## Author Comment (AC1) · 12 Feb 2019

**Response to S. Boersma**

We would like to thank Mr. Boersma for the positive comment regarding the relevance of our work. Mr. Boersma provides a list of additional references and suggests that they should be addressed in the submitted manuscript. We are grateful for the list.

We looked into the proposed references, and indicate here which ones we find relevant for our manuscript:

1) V. Spudic, et al. "Cooperative distributed model predictive control for wind farms", Optimal Control Applications and Methods, 2014.

4) T.N. Jensen, et al. "Fatigue minimising power reference control of a de-rated wind farm" Journal of Physics: Conference Series, 2016.

9) S. Boersma, et al. "A constrained wind farm controller providing secondary frequency regulation: an LES study", Renewable Energy, 2018.

The papers (#1) and (#4) address similar wind farm control problems with different approaches. The paper (#9) became available after the initial submission of our manuscript, and we will consider citing it in our revised manuscript.

We believe that the other suggested papers address aspects which are not extremely relevant to the main contributions of our manuscript:

- The paper (#6) does not include wake interactions in the development and validation of the control algorithm, while the papers (#2) and (#8) consider only laminar flows. On the other hand, our manuscript focuses on practical control solutions for more realistic flow and wake conditions.

- The papers (#3) and (#7) address de-rating strategies for load reduction at the wind turbine level, as opposed to our control concept for the wind farm level.

- The paper (#5) addresses APC requirements for grid connections, which is out of the scope of our submitted manuscript.

---

## Author Comment (AC2) · 12 Feb 2019

**Response to Referee #1**

**Ms Number:** wes-2018-70

**Title:** An active power control approach for wake-induced load alleviation in a fully developed wind farm boundary layer

**Corresponding author:** Mehdi Vali

We would like to thank referee #1 for a thorough review with good suggestions for the improvement of the paper. In the following, we address all the referee's comments. The following table collects the referee's comments, the authors' responses to each point, and the authors' changes in the manuscript. In addition, a color-coded version of the manuscript is provided, in which all changes can be easily identified. We have used the red color to indicate text that has been removed from the submitted manuscript. The descriptions in blue represent the added or re-written parts, addressing the referee's comments.

| Comments of Referee #1 | Authors' Responses |
|---|---|
| (0) One thought I had reading the paper is that perhaps the problem's complexity will grow in a new dimension as additional loads are added, as it is possible that control decisions which help one load, hurt another, and therefore the relative importance weighting between loads will add some complexity (perhaps a necessary arbitrary selection will need to be made, or perhaps some sort of pareto analysis could stand in). Still, this is clearly and properly identified as future work, and so I only comment on it here in case the author's have already thought about this. | **Answer:** We would like to thank the reviewer for this comment. The analysis of the relative importance weighting for structural loading of the individual wind turbines is our current topic of research. We plan to exploit the non-uniqueness of the APC solution in order to balance the lifetime fatigue loading of the individual wind turbines to deal with situations, wherein some wind turbines are highly loaded compared to others. It is more important to consider the different load level/lifetime of the individual turbines rather than several load sensors at each turbine. The latter might only be meaningful if a certain turbine type is suffering from a particularly design problem or a certain turbine component has insufficient design lifetime. Therefore, one possible dimension might be the lifetime fractions, wherein highly loaded wind turbines locally alleviate their own loadings during the APC of wind farms. The implementation of such a wind farm control system is our current research focus. |
| (1) The paper is somewhat long, if it is possible to find opportunities to trim which don't detract too severely from the overall quality, it could be worthwhile | **Answer:** We have revised the manuscript with an effort to trim the text where possible, according to the referee's suggestion. |

| | **Changes in manuscript:** |
|---|---|
| | - In section 3, page 10, line 9, we have shortened and re-written the introductory paragraph as: |
| | "When changing the wind farm power reference, it has to be decided how each turbine should contribute to the power production. The wake interactions among the wind turbines affect the ability the wind farm to track desired power reference trajectories and also increase the aggregated fatigue loading of a wind farm. Therefore, two control objectives are addressed in this study. A closed-loop APC is designed to simultaneously improve the quality of the wind farm power reference tracking and coordinate the power and load distribution in order to influence the wake-induced structural loadings of the wind turbines. " |
| | - In subsection 3.2, page 13, line 4, we have shortened and re-written the introductory paragraph as: |
| | "The wind farm power reference $P^{\text{ref}}$ is distributed among the individual wind turbines on the basis of a power distribution control law, e.g., an open-loop pre-selection (Fleming et al., 2016; van Wingerden et al., 2017) or a closed-loop adjustment (Vali et al., 2018c) of the power set-points. In the current study, we propose an extension to the APC of waked wind farms to actively regulate the distributing set-points, yielding reduced structural loadings when the total power production tracks a time-varying power reference, demanded by the transmission system operator (TSO). The proposed control architecture for APC with a coordinated load distribution (CLD) is depicted in Fig. 7." |
| | - After reviewing the manuscript, we have decided to remove the following text (in subsection 4.1, page 18, line 8) due to similar descriptions in the introductory part of section 4. |
| | "A central open-loop control system, which is studied in Fleming et al. (2016), is considered here as a baseline to share an AGC power signal equally among the wind turbines. Here, the wind farm tracking performance relies on APC at the wind turbine level. The grid stability is degraded when downstream turbines operate either fully or partially inside the wake of their upstream turbines. The closed-loop APC system, presented in subsection 3.2.1, is implemented in PALM coupled with the presented actuator disc models of the wind turbines. The total wind farm tracking error is fed back in order to adjust the equal derating commands against losses caused by the local wake and turbulent effects." |
| (2) The line on page 10 "The wake interactions among the wind turbines play a key role in the stability of the power | **Answer:** We have modified the statement to address the referee's comment. |

| grid" struck me while reading it as perhaps over-stating things. | **Changes in manuscript:** We have modified the statement (page 11, line 2) as follows:

[revised manuscript text omitted]

---

## Author Comment (AC3) · 12 Feb 2019

**Response to Referee #2**

**Ms Number:** wes-2018-70

**Title:** An active power control approach for wake-induced load alleviation in a fully developed wind farm boundary layer

**Corresponding author:** Mehdi Vali

We would like to thank referee #2 for his or her review and comments on our research paper. In the following, we address all the referee's comments. The following table collects the referee's comments, the authors' responses to each point, and the authors' changes in the manuscript. In addition, a color-coded version of the manuscript is provided, in which all changes can be easily identified. We have used the red color to indicate text that has been removed from the submitted manuscript. The descriptions in blue represent the added or re-written parts, addressing the referee's comments.

| Comments of Referee #2 | Authors' Responses |
|---|---|
| (1) The paper is difficult to follow. Nomenclature is not obvious, especially all the subscripts/superscripts. Simple verbal explanations of equations would help. I miss a good concise summary of the specific questions the paper is trying to answer, and the specific ways in which the paper advances the topic. | **Answer:** When writing the manuscript, we did our best to address these issues. We regret that the result was not convincing. We have considered the reviewer's general comments and applied several modifications to improve the revised manuscript. |
| | **Changes in manuscript:**

- We have reviewed the nomenclature and modified the following items:
1) $L$ stands only for the wind turbine loadings,
2) $M_T$ is used instead of $M_{yT}$ to represent the tower base fore-aft bending moment,

Moreover, we have provided the modified nomenclature as an appendix (in page 31) to the revised manuscript with an effort to make it clear for a reader. Additionally, we made sure each equation has been clearly explained.

- In section 1, page 4, line 5, we have reformulated the main contribution of the paper to highlight the main question the paper was trying to answer and advance the topic as:

"The main contribution of this paper is an extension to the APC approaches proposed in Fleming et al. (2016), van Wingerden et al. (2017), and Vali et al. (2018c) to reduce the structural fatigue loading of the individual wind turbines in a waked wind farm by actively coordinating their power set-points, the so-called APC with a coordinated load distribution (CLD) law." |

- In section 4, page 18, line 1, we have added the following text in order to ease following the results and discussion section.

"The remainder of this section is organized as follows. Subsection 4.1 illustrates the wind farm power tracking performance with the Ref. APC, extended with (16), for the cases in which the available wind farm power suddenly drops below the TSO power demand. Subsection 4.2 highlights the performance and the main features of the proposed APC with CLD, compared with the baseline and the Ref. APC. Finally, subsection 4.3 evaluates comparatively the APC with CLD for different wind farm power reference parameterization from some practical criteria, e.g., the power tracking accuracy, the extreme loading, and the structural fatigue loading of the individual wind turbines."

- The conclusion of the paper has been revised. The first paragraph on page 29, line 30 has been modified as follows to summarize briefly the specific achievements in answer to the main question of the paper.

"A new APC approach has been introduced and successfully demonstrated to reduce the wake-induced structural loads of a waked wind farm while the sum of their actual power productions tracks a time-varying wind farm power reference. Since there exist multiple solutions for the APC problem with respect to the distribution of the individual wind turbine power demands, a solution with mitigated structural loading should be possible. The coordinated load distribution law actively adjusts the individual wind turbine power set-points using feedback from the local power and structural load measurements, which makes it practical for real-time control independent of the wind farm size and complexities of wind farm flow and wakes. The study highlights that the enlargement of the APC solution domain increases the controllability of the system for rejecting the wake-induce turbulence effects as the main source of the intensified dynamic loading inside wakes. Indeed, the proposed APC with CLD commands waked wind turbines, subjected to lower wind velocities with higher fluctuations, with realizable power demands using their local power and load measurements. Therefore, their axial induction factors, instead of being saturated to the greedy control setting $a_i$=1/3, gain more freedom to be adjusted for smoother power production and fatigue loading. Moreover, the accuracy of the wind farm power tracking is further improved when the power set-points are chosen according to their locally available power."

| | |
|---|---|
| (2) The paper points out that only tower loads are looked at, which is reasonable as a starting point, but how accurately is this assessed? | **Answer:** We would like to thank the reviewer for this comment. We further emphasize that we are using an extended actuator disc model with a simplified tower model, presented in subsection 2.2, suitable for wind farm control purposes. We have considered the referee's comment to address in the revised manuscript. |
| | **Changes in manuscript:** In subsection 2.2, page 6, line 23, we have added the following text to address the referee's comment.

"In subsection 3.2.2, the tower base fore-aft bending moment is used as representative load indicator to mitigate the wake-induced global load at the different wind turbines inside the wind farm rather than to determine the actual fatigue load damage of the tower or any other component. Therefore the simplified structural and dynamical model (2), (3) and (8) are considered meaningful for our control purpose and applied modeling approach. The ADM turbine model is integrating the turbulence induced loads over the entire swept area and is neglecting important dynamic load effects, e.g., the rotational sampling of partial gusts and partial wakes. If a more realistic rotor model, actuator line model (ALM) with individual blades, is applied, the modeling of the structural dynamics and the load effect of the turbine should be improved as well." |
| (3) The paper gives ambient turbulence intensity of ~5% and waked turbulence intensity of ~15%, but wake turbulence is inhomogeneous, and this does not say anything about the way turbulence varies across and along the wake, or in multiple wakes, nor about the frequency content or length scales of the additional turbulence, all of which will affect tower loads. Does PALM actually deal with all this complexity properly, so that the ~5% and ~15% are just given for interest? | **Answer:** We have considered the valuable comment in the revised manuscript. PALM is an instationary 3D fully-resolved LES code included thermal stratification, and Coriolis forcing. It uses central differences to discretize the non-hydrostatic incompressible Boussinesq approximation of the Navier-Stokes equations on a uniformly spaced Cartesian grid (Maronga et al., 2015). |
| | **Changes in manuscript:**

- In subsection 2.3, page 9, line 5, we have modified the description as follows

"The longitudinal turbulence intensities at hub height of the $1^{st}$ wind turbine are approximately 5% at the turbine location and 15% at 5$D$ distance downstream in the wake center, respectively. These values are calculated after resolving wake structure and its propagation downstream under greedy control as indicators for the assessment of intensified turbulence intensities inside the wake compared with the free stream flow. Indeed, the wake turbulence is inhomogeneous and varies across and along the wake."

- In subsection 2.1, page 5, line 12, we have added the following text: |

| | |
|---|---|
| | "For more information about the general capabilities of the PALM the reader is referred to (Maronga et al., 2015)." |
| (4) Equation (10): How does beta relate to the pitch and torque demands? A given beta can be achieved with different combinations of pitch and torque, and this affects loads. How does this relate to a practical wind turbine controller? | **Answer:** We have modified the manuscript to address the referee's comment. |
| | **Changes in manuscript:** In subsection 3.1, page 11, line 29, we have added the following description to address the referee's comment.

"One important note is that the power de-rating control is governed by the induction control (9). A more comprehensive wind turbine control system would consider the actual torque and pitch controllers. For more details on practical aspects of implementing the torque and pitch control for power de-rating and their effects on the wind turbine loads see (Aho et al., 2013; Aho et al., 2016). |
| (5) End of section 3.2.1: "In such condition, the individual wind turbines should indeed operate at their optimal operating point, i.e., the greedy control": does this ignore the possible (admittedly disputed) potential benefits of wake induction control? | **Answer:** We would like to thank the referee for this comment. We have discussed the operating situation, wherein the total power production temporally falls below the power reference and the wind turbines locally extract the maximum energy from the wind. However, since the focus of the paper is on the power reference tracking instead of power maximization, we do not explore possible power gains through the induction control. Moreover, we consider it unlikely that possible gains during short period when power production drops below the reference could justify the addition control complexity. |
| | **Changes in manuscript:** In subsection 3.2.1, page 14, line 13, we have modified the text to address the referee's comment as:

"In such condition, the individual wind turbines should indeed operate at their optimal operating point, i.e., either the greedy control $a_{i,k} = 1/3$ or the wake induction control (Ciri et al., 2017; Vali et al., 2018a), to extract the maximum amount of energy from the incoming wind." |
| (6) Editors: because the paper is hard to follow, I don't have time to complete the review as thoroughly as I'd like. | **Answer:** We greatly appreciate the time and effort the reviewer has spent on the manuscript. |

[revised manuscript text omitted]